# Chain-of-Zoom: Extreme Super-Resolution via Scale Autoregression and Preference Alignment

**Bryan Sangwoo Kim**    **Jeongsol Kim**    **Jong Chul Ye**
KAIST AI
{bryanswkim, jeongsol, jong.ye}@kaist.ac.kr

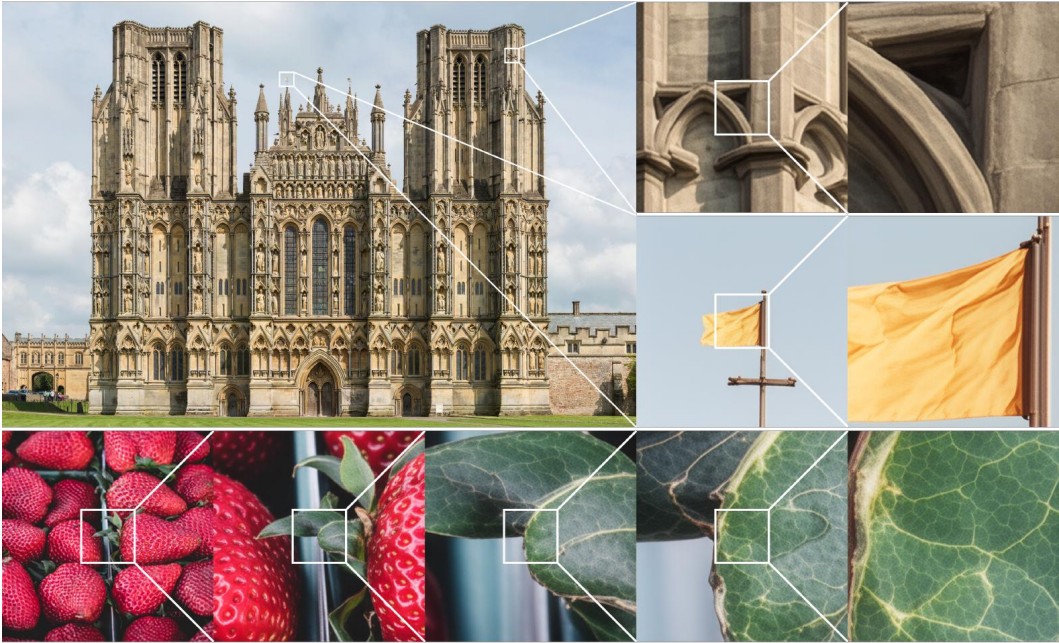

Figure 1: Extreme super-resolution of photorealistic images by CoZ with up to 64× magnification (top) and 256× magnification (bottom). Fine details such as textures on a wall, wrinkles on a flag, and leaf veins are clearly seen.

## Abstract

Modern single-image super-resolution (SISR) models deliver photo-realistic results at the scale factors on which they are trained, but collapse when asked to magnify far beyond that regime. We address this scalability bottleneck with *Chain-of-Zoom (CoZ)*, a model-agnostic framework that factorizes SISR into an autoregressive chain of intermediate scale-states with multi-scale-aware prompts. CoZ repeatedly re-uses a backbone SR model, decomposing the conditional probability into tractable sub-problems to achieve extreme resolutions without additional training. Because visual cues diminish at high magnifications, we augment each zoom step with multi-scale-aware text prompts generated by a vision-language model (VLM). The prompt extractor itself is fine-tuned using Generalized Reward Policy Optimization (GRPO) with a critic VLM, aligning text guidance towards human preference. Experiments show that a standard $4\times$ diffusion SR model wrapped in CoZ attains beyond $256\times$ enlargement with high perceptual quality and fidelity. Project Page: https://bryanswkim.github.io/chain-of-zoom/.

39th Conference on Neural Information Processing Systems (NeurIPS 2025).

# 1 Introduction

The field of generative modeling has witnessed remarkable progress, enabling the synthesis of highly realistic data across various modalities, including images, text, and audio. A key application benefiting from these advancements is single-image super-resolution (SISR), which aims to reconstruct high-resolution (HR) details from a low-resolution (LR) input image. Super-resolution is a problem of core interest for effectively bridging the gap between low-cost imaging sensors and high-fidelity visual information; its usages range from enhancing consumer photographs and legacy media to improving critical details in medical imaging, satellite surveillance, and scientific visualization [2, 29, 31, 41, 44]. The standard approach to SISR is based on the posterior probability distribution:

$$p(\boldsymbol{x}_H \mid \boldsymbol{x}_L) \tag{1}$$

where the goal is to sample a plausible HR image $\boldsymbol{x}_H$ for a given input LR image $\boldsymbol{x}_L$. However, the mapping from $\boldsymbol{x}_L$ to $\boldsymbol{x}_H$ is highly complex and fundamentally ill-posed: a single LR image can correspond to a multitude of plausible HR images. This makes directly modeling the distribution extremely challenging for large magnification factors, and early attempts relying on interpolation or regression often produced blurry results [11, 14, 20, 51]. Recent emergence of powerful generative models (*e.g.*, diffusion-based models) has led to significant advancement in this task, providing strong generative priors over natural images that enable the synthesis of realistic textures and details consistent with the low-resolution input.

Specifically, existing methods leveraging such generative priors largely fall into two categories. One line of work frames SR as an inverse problem, utilizing a pre-trained generative model as a prior during inference time to find a realistic HR image consistent with the LR input [6–9, 21, 22]. While such inverse problem-solving methods benefit from being training-free, they typically require lengthy iterative optimization or sampling processes at inference time to enforce data consistency (*i.e.*, ensuring the downsampled HR prediction matches the original LR input), making them computationally expensive. Another line of work aims to incorporate this data consistency directly into the model's training objective, thereby enabling much faster inference [32, 43, 48, 49, 55, 56]. Modern state-of-the-art models within this category are capable of producing high-quality super-resolved images, even in a single inference step [48, 56].

However, these fast, trained super-resolution models suffer from a significant limitation: they are inherently upper-bounded by their training configuration and tend to collapse when presented with inputs requiring magnification beyond what they were trained on [23, 25, 58]. This failure occurs because the model's internal representations and learned restoration functions are tightly coupled to the specific scale and degradation seen during training [35]. Applying it outside this domain violates its learned assumptions, leading to severe artifacts, blurry outputs, or a complete failure to generate meaningful high-frequency details [12, 14, 20]. This lack of robustness severely restricts the practical applicability of these otherwise powerful models, demanding new models to be trained when the desired magnification factor exceeds what can be currently provided, which is highly inefficient.

In this work, we therefore propose to solve a fundamental question: *How can we effectively utilize super-resolution models to explore much higher resolutions than they were originally trained for?* Solving this question is critical in that it addresses the practical need for flexible and arbitrary-scale super-resolution, allowing users to magnify images to desired levels without being constrained by model training specifics. Furthermore, training models for extremely high magnification factors (*e.g.*, 16x, 32x) directly is often computationally prohibitive due to memory and time constraints [46]. Enabling the extension of existing, well-trained models (*e.g.*, 4x SR models) to higher factors offers a significantly more resource-efficient pathway to achieving extreme resolutions.

To address these fundamental challenges, we present *Chain-of-Zoom (CoZ)*, a novel framework for achieving extreme-resolution image generation beyond the training configurations of conventional super-resolution models. Specifically, we introduce intermediate scale-state modeling to bridge the gap between a low-resolution (LR) input and a high-resolution (HR) target image. These intermediate scale-states enable the decomposition of the conditional distribution in Eq. (1) into a series of tractable components, forming the basis of a scale-level autoregressive (AR) framework. Within this framework, models can progressively generate high-quality images at resolutions previously considered unattainable. In particular, building on the scale-level AR-2 model, we further propose a multi-scale-aware prompt extraction technique. This approach leverages Vision-Language Models (VLMs) to extract descriptive text prompts by attending to *multiple* scale-states throughout the

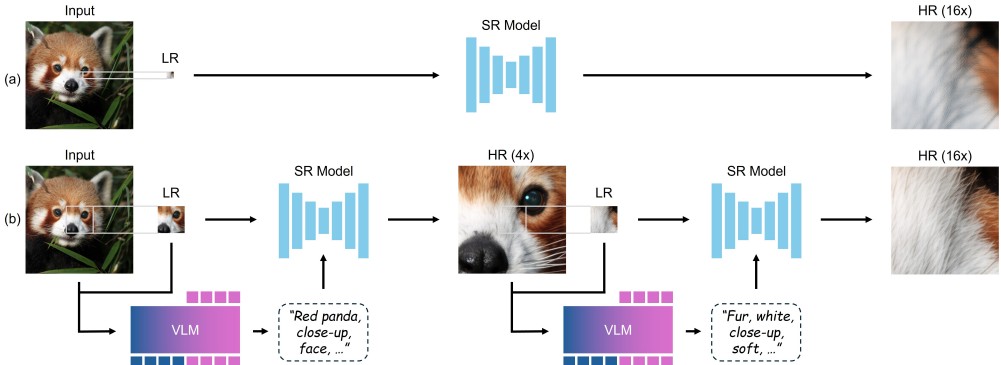

Figure 2: **(a) Conventional SR.** When an SR backbone trained for a fixed up-scale factor (*e.g.*, 4×) is pushed to much larger magnifications beyond its training regime, blur and artifacts are produced. **(b) Chain-of-Zoom (ours).** Starting from an LR input, a pretrained VLM generates a descriptive prompt, which—together with the image—is fed to the same SR backbone to yield the next HR scale-state. This prompt-and-upscale cycle is repeated, allowing a single off-the-shelf model to climb to extreme resolutions (16×–256×) while preserving sharp detail and semantic fidelity.

zooming process, enabling semantically aligned and coherent super-resolution. This is from the observation that at extreme resolutions, conditioning provided by the original signal $x_L$ becomes insufficient, thus leading to unreasonable hallucinations by the SR model in cases.

Furthermore, to obtain text prompts of even richer detail that aligns with human preference, we fine-tune the prompt-extraction VLM under a novel RLHF pipeline leveraging GRPO [34]. A core part of this pipeline is the utilization of a critic VLM to score the outputs of the prompt extraction VLM, thus guiding it to produce prompts more aligned to human preference. Incorporated into the CoZ framework, our final VLM model successfully guides the super-resolution process towards reasonable high-quality results.

In summary, our contributions are as follows:

- We present *Chain-of-Zoom*, a scale-level autoregressive framework that decomposes super-resolution into a sequence of intermediate scale-states and multi-scale-aware prompts, enabling any existing SR model to reach much higher magnifications without retraining.

- We propose a novel RL pipeline for tuning prompt-extraction VLMs with GRPO. This pipeline incorporates appropriate reward functions and a critic reward model to endue multi-scale aware reasoning capabilities to the prompt-extraction VLM.

## 2 Related Work

**Multi-Scale Image Generation and Super-Resolution.** Unconditional multi-scale generators synthesize ever-larger images by passing coarse outputs through successive refinement stages. Cascaded Diffusion Models [17] pioneer this coarse-to-fine pipeline, while AnyresGAN [3], Scalespace-GAN [47], Generative Powers of Ten [45], ZoomLDM [53], and Make-a-Cheap-Scaling [16] share weights across latent zoom levels to reach megapixel resolutions. Because they are generation-based, these methods do not enforce consistency with a given low-resolution input. For true SR, PULSE [27] searches a GAN latent space, and Zoomed In, Diffused Out [28] alternates diffusion denoising with explicit up-sampling, but both do not explore extreme resolutions as in this work.

**Autoregressive Factorizations.** Classic autoregressive models such as PixelCNN, PixelRNN [39, 40] and VAR [38] predict spatial tokens sequentially within a fixed resolution. Pixel Recursive SR [10] extends this to super-resolution by autoregressing over *pixels* after each enlargement—effective for small factors but computationally prohibitive at extreme scales. The proposed CoZ instead autoregresses over *scale-states*: we factorize $p(x_H \mid x_L)$ into a tractable sequence of intermediate zoom distributions, enabling arbitrarily high magnifications without retraining at every factor.

**Diffusion-Based Super-Resolution.** Diffusion models have become the de-facto approach for high-fidelity SISR. SR3 [32] first denoised noisy HR guesses into realistic outputs with diffusion models. StableSR [43] reuses a diffusion prior for faster convergence, and prompt-aware variants (*e.g.*, SeeSR [49], SUPIR [55]) add textual conditioning to bolster semantic faithfulness. OSEDiff [48] distills the multi-step chain into a one-step denoising. Because of its accuracy and efficiency, we adopt OSEDiff as the backbone SR module in our CoZ demonstrations. However, CoZ is model-agnostic: the same scaling strategy can wrap any existing text-guided diffusion (or non-diffusion) SR network.

**RL for Vision–Language Guidance.** Reinforcement learning with human feedback (RLHF) is now widely used to align VLM behaviour with user preference. Early vision-grounded efforts such as LLaVA-RLHF [36] and LLaVACritic [50] employ reward models or critic networks to refine image-conditioned dialogue. Generalized Reward Policy Optimization (GRPO) was introduced by Shao et al. [34] as a policy-space alternative to PPO [33]. GRPO has since been adopted in vision tasks outside SR: Seg-Zero [26] uses GRPO to train VLMs for open-set semantic segmentation, while MetaSpatial [30] applies it to 3-D spatial reasoning in virtual environments. Building on these precedents, we are the first to bring GRPO to prompt-extraction in super-resolution. Our pipeline fine-tunes a prompt-extraction VLM with a composite reward objective unexplored in prior SR work.

## 3 Chain-of-Zoom

### 3.1 Intermediate Scale-State Modeling

In the CoZ framework, we propose to bridge the gap between a target HR image $x_H \in \mathbb{R}^{d_n}$ and an input LR image $x_L \in \mathbb{R}^{d_0}$ by introducing intermediate scale-states $x_i \in \mathbb{R}^{d_i}$. Suppose that an image generative process is modeled as a sequence $(x_0, x_1, ..., x_n)$ where $x_0 := x_L$, $x_n := x_H$, and consecutive states have dimension ratio $s$ (i.e. $d_i = s d_{i-1}$) larger than 1. Under the Markov assumption, the joint distribution could be modeled as $p(x_0, x_1, ..., x_n) = p(x_0) \prod_{i=1}^{n} p(x_i | x_{i-1})$. However, if the model follows a Markov chain structure, relying solely on the transition probability $p(x_i | x_{i-1})$ leads to loss of high-frequency details as $n$ increases (see Fig. 3). Inspired by recent work in inverse problems [8, 21] that demonstrate the effectiveness of text embeddings in reducing the solution space and improving super-resolution between consecutive scales, we therefore introduce latent variables $c_i$ through text embeddings. The text prompt extraction supplements information of the overall zoom process.

Importantly, to reduce hallucinations caused by incorrect text guidance across scale, we find that multi-scale aware text extraction is necessary by feeding $x_{i-1}$ *and* the coarser state $x_{i-2}$ in prompt generation, leading to the conditional probability for the prompt:

$$p_\phi(c_i \mid x_{i-1}, x_{i-2}). \tag{2}$$

Therefore, instead of using the Markov assumption, we propose AR-2 modeling of the image generative process with multi-scale-aware prompts as latent variables:

$$p(x_0, x_1, ..., x_n) = p(x_0, x_1) \prod_{i=2}^{n} p(x_i | x_{i-1}, x_{i-2}), \tag{3}$$

$$p(x_i | x_{i-1}, x_{i-2}) = \int p(x_i | x_{i-1}, x_{i-2}, c_i) p(c_i | x_{i-1}, x_{i-2}) dc_i. \tag{4}$$

Then, the joint distribution of the sequence $(x_0, c_1, x_1, ..., c_n, x_n)$ is expressed as follows:

**Proposition 1.** *Given a sequence of scale-states $x_i$ that follows a AR-2 structure and latent variables $c_i$ that satisfy Eq. (2), the joint distribution is expressed as*

$$p(x_0, c_1, x_1, ..., c_n, x_n) = p(x_0, c_1, x_1) \prod_{i=2}^{n} p(x_i | x_{i-1}, x_{i-2}, c_i) p(c_i | x_{i-1}, x_{i-2}). \tag{5}$$

Now, our objective function is maximizing the likelihood of the entire joint distribution of $x_i$ and $c_i$. Taking the logarithm of Eq. (5), we get the objective function to be maximized:

$$\mathcal{L} = \underbrace{\log p(x_0, c_1, x_1)}_{\mathcal{L}_{\text{init}}} + \underbrace{\sum_{i=2}^{n} \log p(x_i | x_{i-1}, x_{i-2}, c_i)}_{\mathcal{L}_{\text{SR}}} + \underbrace{\sum_{i=2}^{n} \log p(c_i | x_{i-1}, x_{i-2})}_{\mathcal{L}_{\text{VLM}}} \tag{6}$$

where $\mathcal{L}_{\text{init}}$ represents the initial super-resolution step.

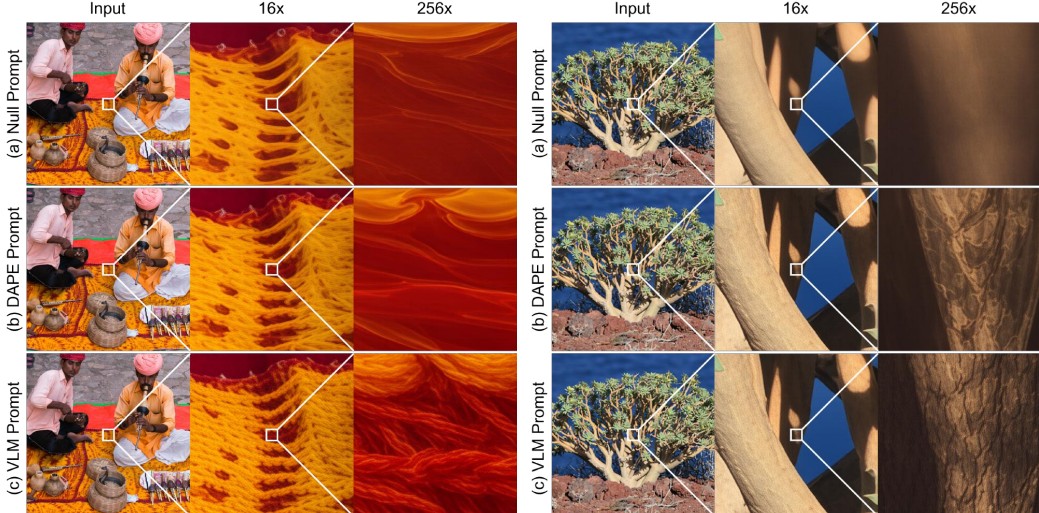

Figure 3: **Significance of proposed multi-scale-aware prompts:** *(a) Null prompt*: coarse structure is retained, but high-frequency details are smoothed out. *(b) DAPE prompt*: inserting text from a degradation-aware prompt extractor (DAPE) helps, yet the images lack intricate detail at large magnifications. *(c) VLM-generated prompts (ours)*: multi-scale prompts extracted by a VLM steer the SR backbone to synthesize realistic textures and crisp details.

## 3.2 Training Objective

The additive structure of the components in Eq. (6) allows for the independent optimization of each term. We achieve this via **next $x_i$ prediction** and **next $c_i$ prediction**, using parameterized models $\theta$ and $\phi$, respectively.

**Next $x_i$ prediction.** The training objective $\mathcal{L}_{\text{SR}}$ represents the likelihood of $x_i$ given previous scale-states $x_{i-1}, x_{i-2}$ and description $c_i$ for $x_i$. Under the assumption that the distribution $p(x_i|x_{i-1}, x_{i-2}, c_i) := \mathcal{N}(x_i; f_\theta(x_{i-1}, x_{i-2}, c_i), \sigma^2 \mathbf{I})$ is Gaussian, where the parameterized model $f_\theta$ predicts the conditional mean of the distribution, the likelihood of $x_i$ is equivalent to

$$\log p(x_i|x_{i-1}, x_{i-2}, c_i) = -\frac{1}{2\sigma^2}\|x_i - f_\theta(x_{i-1}, x_{i-2}, c_i)\|^2 + C \tag{7}$$

where $C = -\frac{d_i}{2}\log(2\pi\sigma^2)$. To reduce the computational complexity of training $f_\theta$, our key idea is that its dependency to $x_{i-2}$ is only through the multi-scale-aware prompt, i.e. $c_i = c_i(x_{i-1}, x_{i-2})$, leading to $f_\theta(x_{i-1}, x_{i-2}, c_i) = f_\theta(x_{i-1}, c_i(x_{i-1}, x_{i-2}))$. Maximizing the simplified likelihood thus reduces to minimizing the mean-squared error (MSE) between the predicted HR patch from $x_{i-1}$ and the ground truth—precisely the loss most SR backbones are already trained with. In this work, we perform experiments with a backbone SR model trained via settings in Sec. 4.1, yet our framework is model-agnostic.

**Next $c_i$ prediction.** Recall that the dependency to the $x_{i-2}$ in AR-2 model is through the multi-scale aware prompt extraction, which supplements information of the overall zoom process and reduces hallucinations caused by incorrect text guidance. For a single zoom step $i$, the prompt $c_i = (c_{i,1}, \cdots, c_{i,T_i})$ is a token sequence conditioned on the current and previous image, i.e. $x_{i-1}, x_{i-2}$. Modern VLMs model this distribution autoregressively:

$$p_\phi(c_i \mid x_{i-1}, x_{i-2}) = \prod_{t=1}^{T_i} p_\phi(c_{i,t} \mid x_{i-1}, , x_{i-2}, c_{i,<t}) \tag{8}$$

where $c_{i,<t} = (c_{i,1}, \cdots, c_{i,t-1})$. Maximizing the log-likelihood $\log p_\phi(c_i \mid x_{i-1}, x_{i-2})$ therefore amounts to minimizing the negative log-likelihood (cross-entropy) for each token:

$$\mathcal{L}_{\text{VLM}}^{(i)} = -\log p_\phi(c_i \mid x_{i-1}, x_{i-2}) = -\sum_{t=1}^{T_i} \log p_\phi(c_{i,t} \mid x_{i-1}, x_{i-2}, c_{i,<t}). \tag{9}$$

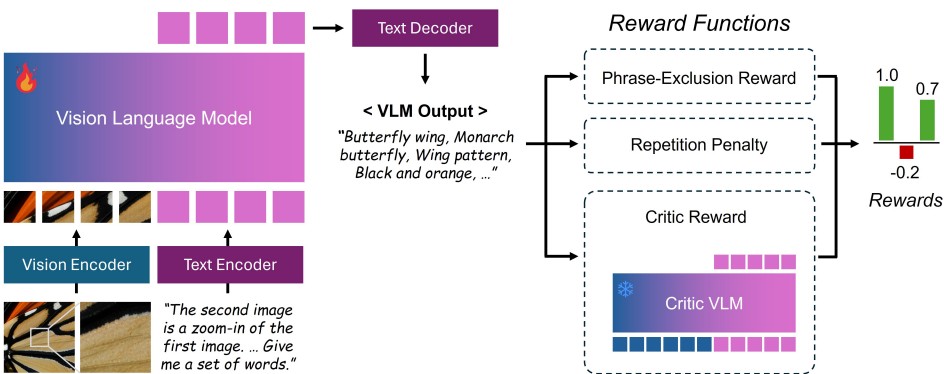

Figure 4: **GRPO Training Framework.** At every zoom step, multi-scale image crops are fed to the base VLM, which generates candidate prompts after perceiving input images. A critic VLM scores the prompt for semantic quality, while phrase-exclusion and repetition penalties enforce conciseness and relevance. The weighted sum of these rewards forms the GRPO signal that iteratively fine-tunes the base VLM, steering it towards prompts that best guide extreme-scale super-resolution.

Eq. (9) is exactly the standard next-token cross-entropy loss used to pre-train modern VLMs; hence our framework can employ any off-the-shelf VLM whose weights already maximize this objective.

**Inference.** Given pre-trained parameterized models $\theta$ and $\phi$, the sequence ($x_0$, $c_1$, $x_1$, ..., $c_n$, $x_n$) can be generated recursively. Starting from the low-resolution image $x_L = x_0$, a description for the next scale, $c_1 \sim p_\phi(c_1 \mid x_0)$, is first sampled. Then, the next scale state is generated by sampling $x_1 \sim p_\theta(x_1 \mid x_0, c_1)$. For subsequent steps, the description at scale $i$ is sampled as $c_i \sim p_\phi(c_i \mid x_{i-1}, x_{i-2})$, followed by sampling the image at that scale as $x_i \sim p_\theta(x_i \mid x_{i-1}, x_{i-2}, c_i)$. This sequential sampling process generates specific, plausible high-resolution outputs $x_n$ without needing to model the full marginal distribution $p(x_0, ..., x_n)$ explicitly. When using SR backbone models that require input and output dimensions to be identical (*e.g.*, Stable-diffusion-based SR models [43, 48, 49, 55]), a fixed-size window is cropped from the HR image and resized to the required dimension. Thus, super-resolution operates in local regions, and achieving outputs of entire images would require multiple runs of CoZ.

### 3.3 Training Multi-Scale-Aware Prompt Extraction using RL

At extreme magnification factors, the visual evidence in the input image becomes extremely sparse, causing the SR backbone model to rely more heavily on text prompts. To curb the ensuing drift towards implausible high-frequency hallucinations, we fine-tune the prompt-extraction VLM so that its textual guidance aligns with human aesthetic and semantic preferences. Our fine-tuning pipeline (Fig. 4) adopts Generalized Reward Policy Optimization (GRPO). For each zoom step $i$, the VLM receives multi-scale image crops ($x_{i-2}, x_{i-1}$) and produces a candidate prompt $c_i$. The prompt is scored by a set of task-specific reward functions, and the weighted sum $R(c_i)$ drives the GRPO update to align the VLM prompts with human preference. The overall reward $R(c_i)$ is a weighted sum of three components, each targeting a distinct failure mode observed during preliminary experiments:

$$R(c_i) = w_{\text{critic}} R_{\text{critic}} + w_{\text{phrase}} R_{\text{phrase}} + w_{\text{rep}} R_{\text{rep}} \qquad (10)$$

**Critic Preference Reward ($R_{\textbf{critic}}$).** A stronger vision–language critic VLM judges the candidate prompt in the context of the input multi-scale image crops and assigns a raw score in $[0, 100]$. We linearly rescale this score to $[0, 1]$ and treat it as a proxy for human preference, thereby imbuing the prompt-extraction VLM with the critic VLM's higher-level semantic priors.

**Phrase-Exclusion Reward ($R_{\textbf{phrase}}$).** Multi-image conditioning occasionally leads the prompt-extraction VLM to emit viewpoint markers such as "*first image*" or "*second image*," which are meaningless to the downstream SR model. We therefore issue a reward of 1 if none of a predefined blacklist of such phrases appear, and 0 otherwise.

**Repetition Penalty ($R_{\textbf{rep}}$).** Following Yeo et al. [54], we compute the fraction of repeated $n$-grams in the prompt and give a negative reward (down to $-1$) for a higher repetition ratio.

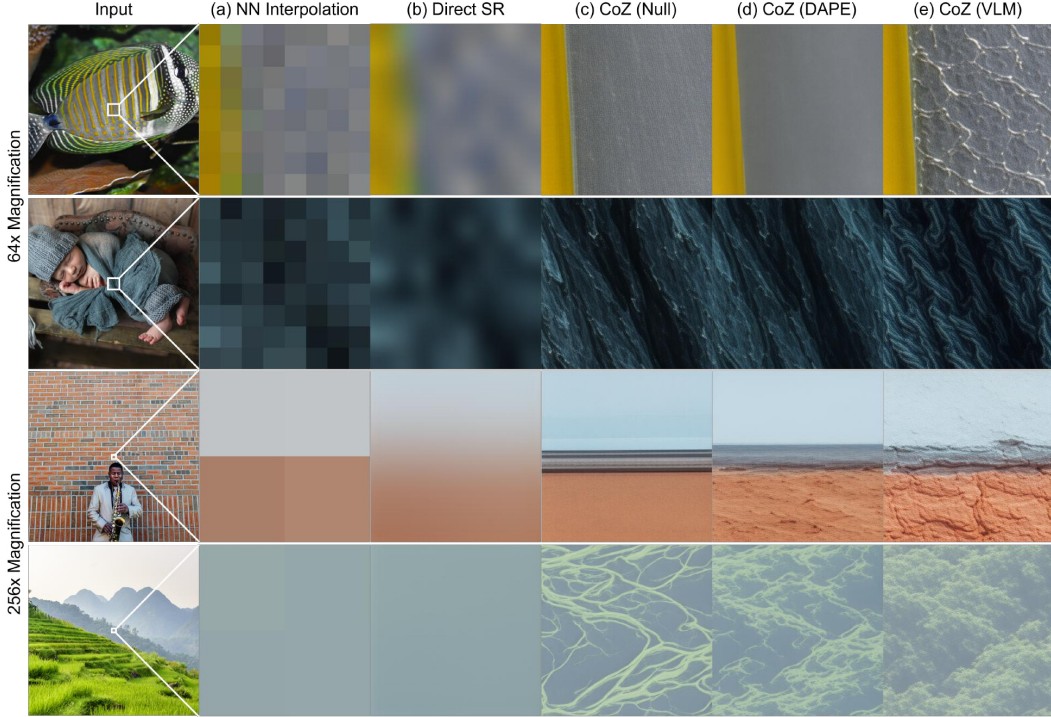

Figure 5: **Qualitative Results.** For each input image, super-resolution is performed on different magnifications with various methods: **(a) Nearest neighbor interpolation**; **(b) One-step direct SR** with the backbone SR model; **(c-e) Variants of CoZ** with different text prompts. The CoZ framework shows significantly better performance at large magnifications. Furthermore, using CoZ with VLM prompts assists the SR model in generating realistic details without hallucinations.

## 4 Experiments

### 4.1 Experimental Settings

We adopt the setup of prior work [49, 48] and train OSEDiff [48] as the backbone SR model with the LSDIR [24] dataset and 10K images from FFHQ [18]. We use Stable Diffusion 3.0 [13] as the backbone diffusion model and adopt a coarse-to-fine training strategy: first training on random degradation, and then training specifically for $4\times$ magnifications. Text guidance is provided by Degradation-Aware Prompt Extractor (DAPE) [49] as the naive prompt extractor, while Qwen2.5-VL-3B-Instruct [37] is used as the prompt-extraction VLM. RLHF training with GRPO is performed with InternVL2.5-8B [5] as the critic VLM. The same dataset used for training the backbone SR model is also used for GRPO training, and weights are given as: $w_{\text{critic}} = 1.0$, $w_{\text{phrase}} = 0.5$, $w_{\text{rep}} = 0.5$.

Evaluation is performed on the training datasets of DIV2K [1] and DIV8K [15], consisting of 800 images and 1500 images, respectively. Each image is resized and center-cropped to resolution of $512 \times 512$ to be input to the SR model. For four recursions, the HR image of the previous zoom is center-cropped and resized by a scale of 4 back to the resolution of $512 \times 512$.

### 4.2 Comparison Results

We perform comparison across four recursions for various methods. Specifically, we compare between nearest neighbor interpolation, direct magnification via one-step SR, and three versions of the proposed CoZ leveraging different prompts (*i.e.*, Null, DAPE, VLM).

**Qualitative Comparison.** Qualitative results in Fig. 5 show that nearest neighbor interpolation and one-step direct SR fall off at higher scales, while CoZ variants produce images of better quality. Thus, incorporating VLM prompts helps overcome the sparsity of the original input signal.

Table 1: Quantitative comparison on no-reference metrics. **Bold**: best, Underline: second-best.

| Scale | Method | DIV2K | | | | DIV8K | | | |
|---|---|---|---|---|---|---|---|---|---|
| | | NIQE↓ | MUSIQ↑ | MANIQA↑ | CLIPIQA↑ | NIQE↓ | MUSIQ↑ | MANIQA↑ | CLIPIQA↑ |
| 4× | NN Interpolation | 12.1252 | 39.96 | 0.3396 | 0.2630 | 13.1984 | 40.26 | 0.3472 | 0.2672 |
| | Direct SR | 4.7320 | 67.00 | 0.6344 | 0.7005 | 4.8631 | 66.29 | 0.6359 | 0.6946 |
| | CoZ (Null) | 4.7706 | 66.99 | 0.6309 | 0.6977 | 4.9011 | 66.23 | 0.6325 | 0.6897 |
| | CoZ (DAPE) | 4.7312 | 67.01 | 0.6344 | 0.7004 | 4.8607 | 66.29 | 0.6359 | 0.6946 |
| | CoZ (VLM) | **4.6572** | **67.10** | **0.6360** | **0.7017** | **4.8099** | **66.37** | **0.6370** | **0.6953** |
| 16× | NN Interpolation | 22.1215 | 24.01 | 0.3378 | 0.2346 | 22.2744 | 24.94 | 0.3465 | 0.2585 |
| | Direct SR | 7.2183 | 51.25 | 0.5406 | 0.6080 | 7.5855 | 50.17 | 0.5473 | 0.6035 |
| | CoZ (Null) | 6.5016 | **59.19** | 0.5859 | **0.6686** | 6.7898 | **58.04** | 0.5881 | 0.6618 |
| | CoZ (DAPE) | 6.5456 | 58.83 | 0.5946 | 0.6609 | 6.8607 | 57.79 | 0.5964 | **0.6628** |
| | CoZ (VLM) | **6.3957** | 58.81 | **0.5970** | 0.6574 | **6.6500** | 57.99 | **0.6006** | 0.6615 |
| 64× | NN Interpolation | 27.4051 | 37.69 | 0.3803 | 0.3690 | 27.7533 | 37.13 | 0.3861 | 0.3837 |
| | Direct SR | 16.5915 | 22.54 | 0.3995 | 0.4309 | 16.5874 | 22.97 | 0.4069 | 0.4451 |
| | CoZ (Null) | 8.3500 | 51.82 | 0.5627 | 0.6305 | 8.5694 | 50.96 | 0.5638 | 0.6240 |
| | CoZ (DAPE) | 8.6598 | 51.77 | 0.5726 | 0.6262 | 8.7669 | 50.40 | 0.5714 | 0.6274 |
| | CoZ (VLM) | **8.2335** | **52.13** | **0.5788** | **0.6315** | **8.2992** | **51.20** | **0.5787** | **0.6282** |
| 256× | NN Interpolation | 34.8461 | 27.01 | 0.4179 | 0.5259 | 37.2612 | 26.98 | 0.4184 | 0.5299 |
| | Direct SR | 16.1749 | 28.89 | 0.4470 | 0.5196 | 15.8667 | 28.90 | 0.4464 | 0.5256 |
| | CoZ (Null) | 10.0456 | 46.28 | 0.5510 | 0.5857 | 10.0630 | 46.56 | 0.5479 | 0.5899 |
| | CoZ (DAPE) | 10.4569 | 46.22 | 0.5564 | 0.5889 | 10.2788 | 45.81 | 0.5535 | 0.5984 |
| | CoZ (VLM) | **9.8260** | **47.83** | **0.5692** | **0.5986** | **9.6405** | **47.25** | **0.5646** | **0.6041** |

**Quantitative Comparison.** Quantitative results are given in Tab. 1. Due to the non-availability of ground-truth images for 256× magnifications, we follow [27] and evaluate performance on no-reference perceptual metrics. Specifically, we use the metrics NIQE [57], MUSIQ [19], MANIQA-pipal [52], CLIPIQA [42] for a thorough evaluation. At low scales (*i.e.*, Scale 4×), difference between methods is minimal, but at high scales (*i.e.*, Scales 64×, 256×) the proposed framework shows consistently better performance. Furthermore, prompts by DAPE show comparable performance at low scales but fall off at higher scales, while VLM-generated prompts exhibit significantly better performance, supporting our claim that prompt-extraction by VLMs make up for the deficient visual conditioning provided by the initial image.

**Quantitative Comparison with Baseline Methods.** In Tab. 2, we further provide quantitative comparison with baseline methods: arbitrary-scale SR methods (LIIF [4]) and direct super-resolution of diffusion-based SR methods (SeeSR [49], S3Diff [56]). All three baselines show greatly degraded performance at high magnifications. For the case of S3Diff, we additionally show quantitative results for applying CoZ to the pretrained, freely accessible S3Diff, leveraging our multi-scale aware GRPO fine-tuned VLM. All results clearly confirm the significant performance improvement by CoZ. Additional results for performing CoZ with OSEDiff leveraging the Stable Diffusion v2.1 backbone is provided in Appendix E.

## 4.3 GRPO for VLM

**GRPO Training.** Reward graphs for training the prompt-extraction VLM with different critic VLMs are shown in Fig. 6 and Fig. 7. When using InternVL2.5-8B [5] as the critic VLM, phrase exclusion reward and repetition penalty converge to 1.00 and 0.00 (respectively) in the early stages of training, while the critic reward increases gradually throughout the training process. Similar trends are observed when using Qwen2.5-VL-7B-Instruct [37] as the critic VLM, proving the robustness of our method. Additional quantitative results for this case is provided in Appendix F.

**Preference Alignment.** Using an off-the-shelf VLM for prompt-extraction can cause unwanted hallucinations to occur in the zoom process. An example case is shown in Fig. 8 (Top), where the off-the-shelf VLM generates improper prompts due to insufficient knowledge of the initial image at high magnifications. By inducing the VLM to generate multi-scale-aware prompts by conditioning on $(x_{i-1}, x_{i-2})$, we can produce more suitable prompts Fig. 8 (Middle). Finally, using the VLM fine-tuned with GRPO we can produce high-quality samples while reducing unwanted hallucinations as in Fig. 8 (Bottom). We further prove that the VLM after undergoing GRPO training is better aligned with human preference through user study. For this, we follow prior work [27], and perform a MOS (mean-opinion-score) test on various samples. Results and details are included in Appendix D.

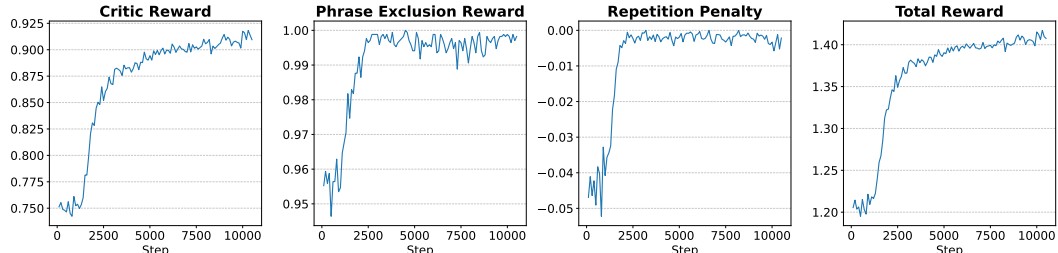

Figure 6: Reward graphs of using **InternVL2.5-8B** as the critic VLM, evaluated on a validation set. Values for Critic Reward, Phrase Exclusion Reward, Repetition Penalty, and Total Reward increase throughout the training process.

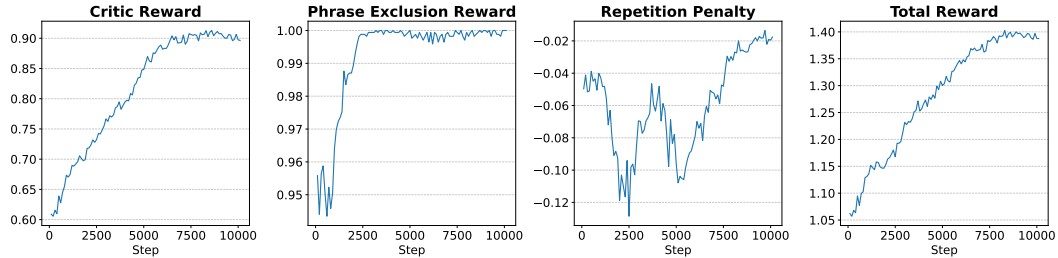

Figure 7: Reward graphs of using **Qwen2.5-VL-7B-Instruct** as the critic VLM, evaluated on a validation set. Values for Critic Reward, Phrase Exclusion Reward, Repetition Penalty, and Total Reward increase throughout the training process.

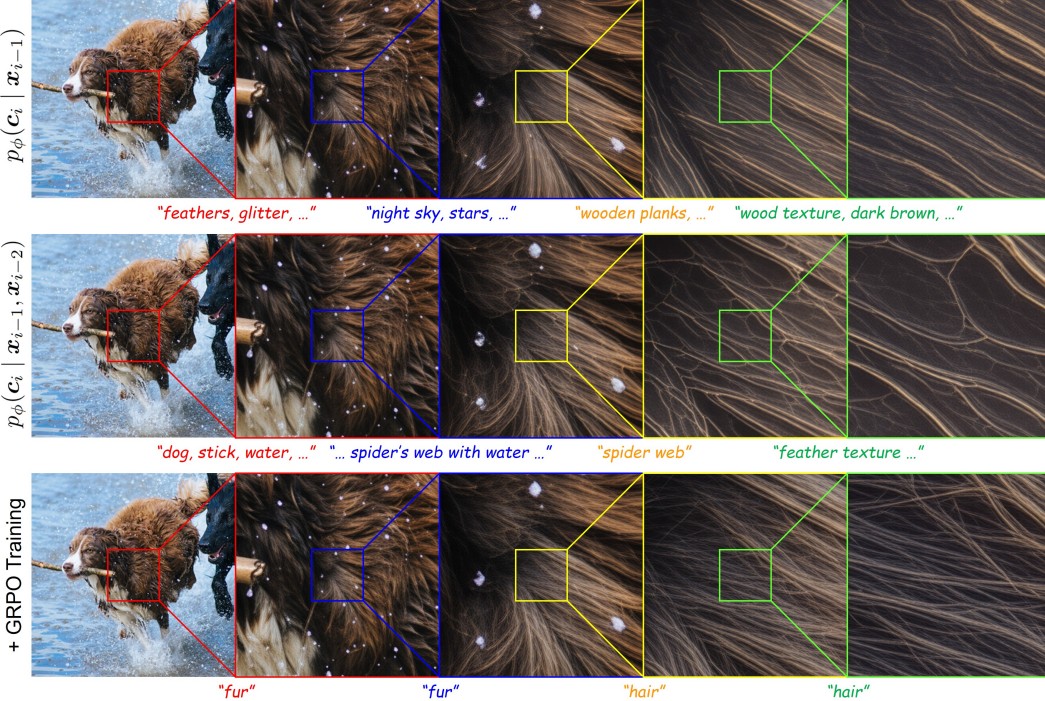

Figure 8: RLHF training with GRPO assists the prompt-extraction VLM in creating meaningful prompts for accurate guidance. (Top) **Base VLM**: generating prompts only from the LR input causes unwanted hallucinations as shown by the incorrect prompts; (Middle) **Multi-scale image prompts** are helpful at low scales (*e.g.*, accurate prompt of "dog, stick, water, ...") but fail at high scales; (Bottom) **VLM aligned with human preference** guides samples with improved text guidance.

Table 2: Quantitative comparison with baseline methods. **Best**, Second-Best.

| Scale | Method | DIV2K | | | | DIV8K | | | |
|---|---|---|---|---|---|---|---|---|---|
| | | NIQE↓ | MUSIQ↑ | MANIQA↑ | CLIPIQA↑ | NIQE↓ | MUSIQ↑ | MANIQA↑ | CLIPIQA↑ |
| 4× | LIIF | 6.6210 | 57.47 | 0.5456 | 0.4587 | 6.8050 | 55.52 | 0.5386 | 0.4545 |
| | SeeSR | 5.5208 | 57.56 | 0.5387 | 0.5535 | 5.5940 | 55.50 | 0.5346 | 0.5385 |
| | S3Diff | 4.9803 | 65.82 | 0.6361 | 0.6596 | 5.0305 | 64.08 | 0.6328 | 0.6481 |
| | S3Diff + CoZ | **4.8637** | **67.18** | **0.6459** | **0.6835** | **4.9414** | **65.31** | **0.6405** | **0.6680** |
| 16× | LIIF | 11.4815 | 25.94 | 0.2860 | 0.3024 | 11.8734 | 26.73 | 0.2896 | 0.3178 |
| | SeeSR | 9.6798 | 41.68 | 0.4310 | 0.4669 | 9.8865 | 40.02 | 0.4210 | 0.4601 |
| | S3Diff | 6.7383 | 51.14 | 0.5305 | 0.5886 | 7.2399 | 50.32 | 0.5370 | 0.5841 |
| | S3Diff + CoZ | **6.7310** | **56.82** | **0.5752** | **0.6168** | **6.8698** | **56.46** | **0.5836** | **0.6218** |
| 64× | LIIF | 16.9215 | 20.02 | 0.3451 | 0.4131 | 17.4912 | 20.43 | 0.3522 | 0.4250 |
| | SeeSR | 16.9095 | 21.83 | 0.4106 | 0.4193 | 16.9275 | 22.68 | 0.4167 | 0.4309 |
| | S3Diff | 16.1421 | 21.54 | 0.3893 | 0.4917 | 16.5506 | 22.00 | 0.3968 | 0.5089 |
| | S3Diff + CoZ | **8.7770** | **48.90** | **0.5490** | **0.5801** | **8.9794** | **48.07** | **0.5560** | **0.5909** |
| 256× | LIIF | 23.8949 | 26.05 | 0.4108 | 0.5380 | 24.3300 | 26.01 | 0.4116 | 0.5423 |
| | SeeSR | 20.9635 | 25.81 | 0.4438 | 0.5193 | 19.9628 | 25.94 | 0.4429 | 0.5203 |
| | S3Diff | 16.7809 | 25.92 | 0.4324 | 0.5359 | 16.9952 | 25.91 | 0.4312 | 0.5415 |
| | S3Diff + CoZ | **10.7668** | **43.59** | **0.5438** | **0.5570** | **10.5001** | **43.31** | **0.5417** | **0.5673** |

Table 3: Runtime analysis for different methods (seconds).

| Scale | Phase | Direct SR (DAPE) | CoZ (Null) | CoZ (DAPE) | CoZ (VLM) |
|---|---|---|---|---|---|
| 4× | SR | 0.1467 | 0.1443 | 0.1460 | 0.1445 |
| | PE | 0.0136 | 0.0000 | 0.0130 | 0.3777 |
| 16× | SR | 0.1462 | 0.2886 | 0.2912 | 0.2896 |
| | PE | 0.0130 | 0.0000 | 0.0254 | 0.7068 |
| 64× | SR | 0.1462 | 0.4329 | 0.4363 | 0.4349 |
| | PE | 0.0131 | 0.0000 | 0.0382 | 1.0301 |
| 256× | SR | 0.1462 | 0.5774 | 0.5816 | 0.5802 |
| | PE | 0.0132 | 0.0000 | 0.0509 | 1.3505 |

## 4.4 Runtime Analysis

Runtime analysis for direct super-resolution and CoZ variants across varying scales ($4\times$ to $256\times$) is given in Tab. 3. The average inference time required (in seconds) to apply CoZ on a single image is evaluated on 500 images of the DIV2K dataset. We divide inference into two phases: super-resolution (SR) and prompt extraction (PE); computational time required for each phase is analyzed accordingly.

## 5 Conclusion

This paper tackles the long-standing scalability gap in single-image super-resolution: state-of-the-art models excel at their trained scale factors yet fail when asked to enlarge images far beyond that range. Specifically, we introduced *Chain-of-Zoom (CoZ)*, a scale-level autoregressive framework that transforms any existing SR backbone into an extreme-magnification engine by decomposing the LR to HR mapping into a sequence of intermediate scale-states and multi-scale-aware prompts. CoZ is model-agnostic, requires no retraining of the base network, and thus offers a cost-effective path up to extreme resolutions. In particular, to maintain semantic coherence as visual evidence thins out, we leverage a multi-scale-aware prompt extractor driven by a VLM fine-tuned through a GRPO-based RLHF pipeline. Overall, CoZ yields sharp, realistic results at extreme scales while keeping inference efficient. By decoupling super-resolution performance from fixed training magnifications and demonstrating the value of aligned textual guidance, our work opens new avenues for resource-frugal image enhancement and lays a foundation for future exploration of learned zoom policies, domain-specific reward functions, and adaptive backbone selection.

**Limitation and Potential Negative Impacts.** While CoZ enables extreme super-resolution with high visual fidelity, it requires repeated application for extreme magnification, which may cause error accumulation over iterations. Moreover, high-fidelity generation from low-resolution inputs may raise concern regarding misinformation or unauthorized reconstruction of sensitive visual data.

## Acknowledgments and Disclosure of Funding

This work was supported by the National Research Foundation of Korea under Grant RS-2024-00336454, and by the Institute of Information & Communications Technology Planning & Evaluation (IITP) grant funded by the Korean government (MSIT) (No. RS-2024-00457882, AI Research Hub Project). This work was also supported by the IITP (Institute of Information & Communications Technology Planning & Evaluation)-ITRC (Information Technology Research Center) grant funded by the Korea government (Ministry of Science and ICT) (IITP-2025-RS-2020-II201461).

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

# A Proofs

**Proposition 1.** *Given a sequence of scale-states $\boldsymbol{x}_i$ that follows a AR-2 structure and latent variables $\boldsymbol{c}_i$ that satisfy Eq. (2), the joint distribution is expressed as*

$$p(\boldsymbol{x}_0, \boldsymbol{c}_1, \boldsymbol{x}_1, ..., \boldsymbol{c}_n, \boldsymbol{x}_n) = p(\boldsymbol{x}_0, \boldsymbol{c}_1, \boldsymbol{x}_1) \prod_{i=2}^{n} p(\boldsymbol{x}_i | \boldsymbol{x}_{i-1}, \boldsymbol{x}_{i-2}, \boldsymbol{c}_i) p(\boldsymbol{c}_i | \boldsymbol{x}_{i-1}, \boldsymbol{x}_{i-2}). \tag{5}$$

*Proof.* By substituting Eq. (4) to Eq. (3) and expanding the base term $p(\boldsymbol{x}_0, \boldsymbol{x}_1)$, we obtain:

$$\left[ \int p(\boldsymbol{x}_0, \boldsymbol{c}_1, \boldsymbol{x}_1) d\boldsymbol{c}_1 \right] \prod_{i=2}^{n} \left[ \int p(\boldsymbol{x}_i | \boldsymbol{x}_{i-1}, \boldsymbol{x}_{i-2}, \boldsymbol{c}_i) p(\boldsymbol{c}_i | \boldsymbol{x}_{i-1}, \boldsymbol{x}_{i-2}) d\boldsymbol{c}_i \right]$$

$$= \int \cdots \int \left[ p(\boldsymbol{x}_0, \boldsymbol{c}_1, \boldsymbol{x}_1) \prod_{i=2}^{n} p(\boldsymbol{x}_i | \boldsymbol{x}_{i-1}, \boldsymbol{x}_{i-2}, \boldsymbol{c}_i) p(\boldsymbol{c}_i | \boldsymbol{x}_{i-1}, \boldsymbol{x}_{i-2}) \right] d\boldsymbol{c}_1 \cdots d\boldsymbol{c}_n$$

$$= \int \cdots \int p(\boldsymbol{x}_0, \boldsymbol{c}_1, \boldsymbol{x}_1, ..., \boldsymbol{c}_n, \boldsymbol{x}_n) d\boldsymbol{c}_1 \cdots d\boldsymbol{c}_n$$

$$= p(\boldsymbol{x}_0, ..., \boldsymbol{x}_n)$$

This confirms the joint distribution in Eq. (5) is consistent with the marginal AR-2 process. $\quad\square$

# B Experimental Details

## B.1 Model Checkpoints

We use the pretrained VLM models Qwen2.5-VL-3B-Instruct and InternVL2.5-8B, available at `https://huggingface.co/Qwen/Qwen2.5-VL-3B-Instruct` and `https://huggingface.co/OpenGVLab/InternVL2_5-8B`, respectively. We also use the pretrained Stable Diffusion 3.0 model available at `https://huggingface.co/stabilityai/stable-diffusion-3-medium`. Evaluation is performed using the script for testing IQA (Image Quality Assessment) in `https://github.com/cswry/OSEDiff`.

## B.2 User Prompts

The user prompt used for the base VLM is as follows:

> The second image is a zoom-in of the first image. Based on this knowledge, what is in the second image? Give me a set of words.

The user prompt used for the critic VLM is as follows:

> First Image: `<image>`
> Second Image: `<image>`
> The second image is a zoom-in of the first image. Please rate the quality of the following description on how well it describes the second image. Output only a single score between 0 and 100.
> Description: `<Output of Base VLM>`
> Rating (0-100):

## B.3 Other Settings

The backbone SR model is trained based on the training scheme of OSEDiff [48], with Stable Diffusion 3.0 as the backbone diffusion model. We train using four NVIDIA GeForce RTX 3090 GPUs with the LSDIR [24] dataset and 10K images from FFHQ [18]. Coarse-to-fine training is used: random degradation (same setting as OSEDiff) for 25K iterations, then $4\times$ specific upscaling for 20K iterations. Other settings (*e.g.*, batch size, learning rate, etc.) follow the default settings of OSEDiff.

The VLM model is GRPO fine-tuned using four NVIDIA GeForce RTX 3090 GPUs with the LSDIR dataset, with a train/validation split ratio of 0.01 (*i.e.*, 849 images for validation). Specifically, the Qwen2.5-VL-3B-Instruct model is LoRA fine-tuned (Rank: 8, Alpha: 32, Dropout: 0.05), with two generations per prompt for 10K global steps. The SWIFT [59] infrastructure was used for this process. Reward graphs during training for the validation set are given in Fig. 6 of the main paper.

Evaluation is performed with the code provided in [48], modified for no-reference metric evaluation. For occasional failure cases, worst values are given for each metric (100.0 for NIQE, 0.0 for others).

## C Algorithms

The following algorithms are provided:

- Algorithm 1: the main algorithm for Chain-of-Zoom inference.
- Algorithm 2: the algorithm for GRPO-based human preference alignment training of VLMs.

---

**Algorithm 1** Chain-of-Zoom Inference

---

**Input:** Low resolution image $x_L$, Super-resolution model $p_\theta$, VLM $p_\phi$, Number of recursions $n$
**Output:** High resolution image $x_n$
1: $x_0 \leftarrow x_L$
2: **for** $i : 1 \rightarrow n$ **do**
3:     **if** $i = 1$ **then**
4:         $c_i \leftarrow p_\phi(c_i | x_{i-1})$
5:     **else**
6:         $c_i \leftarrow p_\phi(c_i | x_{i-1}, x_{i-2})$
7:     **end if**
8:     $x_i \leftarrow p_\theta(x_i | x_{i-1}, c_i)$
9: **end for**

---

**Algorithm 2** GRPO-based RL Training of Prompt-Extraction VLM

---

**Input:** Base (prompt-extraction) VLM $p_\phi$ with parameters $\phi$, Critic VLM $V_{\text{critic}}$, Phrase blacklist $B_{\text{phrase}}$ for $R_{\text{phrase}}$, Number of training iterations $N_{\text{iter}}$, Number of generations per prompt $N_{\text{gen}}$, Training dataset $D = \{(x_{k-2}^{(j)}, x_{k-1}^{(j)})\}_{j=1}^M$ of multi-scale image crop pairs
**Output:** Fine-tuned prompt-extraction VLM $p_\phi$
1: **for** iteration $t : 1 \rightarrow N_{\text{iter}}$ **do**
2:     **for** generation $g : 1 \rightarrow N_{\text{gen}}$ **do**
3:         Sample a multi-scale image pair $(x_{i-2}, x_{i-1})$ from $D$
4:         Generate candidate prompt $c_i^{(g)} \sim p_\phi(\cdot | x_{i-1}, x_{i-2})$
5:         $s_{\text{critic}} \leftarrow V_{\text{critic}}(c_i^{(g)} | x_{i-1}, x_{i-2})$       ▷ Critic VLM scores prompt, range $[0, 100]$
6:         $R_{\text{critic}} \leftarrow \text{Rescale}(s_{\text{critic}}, 0, 1)$               ▷ Rescale score to $[0, 1]$
7:         $R_{\text{phrase}} \leftarrow 1$
8:         **for all** $b \in B_{\text{phrase}}$ **do**
9:             **if** phrase $b$ is in $c_i^{(g)}$ **then**
10:                 $R_{\text{phrase}} \leftarrow 0$
11:                 **break**
12:             **end if**
13:         **end for**
14:         $R_{\text{rep}} \leftarrow -\text{FractionOfRepeatedNgrams}(c_i^{(g)})$    ▷ Repetition Penalty, range $[-1, 0]$
15:         $R(c_i^{(g)}) \leftarrow w_{\text{critic}} R_{\text{critic}} + w_{\text{phrase}} R_{\text{phrase}} + w_{\text{rep}} R_{\text{rep}}$    ▷ Total weighted reward
16:     **end for**
17:     $\hat{A}^{(g)} \leftarrow R(c_i^{(g)}) - \frac{1}{N_{\text{gen}}} \sum_{n=1}^{N_{\text{gen}}} R(c_i^{(n)})$      ▷ Group-based advantage estimation
18:     Calculate $\mathcal{L}_{\text{GRPO}}(\phi)$ with estimated advantages $\hat{A}^{(g)}$     ▷ Detailed procedure in [34]
19:     Update parameters $\phi$ of $p_\phi$ using GRPO policy update with $\mathcal{L}_{\text{GRPO}}(\phi)$
20: **end for**

# D   User Study

We further prove that GRPO fine-tuning of the VLM enhances human preference alignment by performing a MOS (mean-opinion-score) test on various samples for 25 human participants. Specifically, we compare between three different VLM prompts: (i) prompts generated from only the LR input (*i.e.*, $p_\phi(c_i \mid x_{i-1})$); (ii) prompts generated from multi-scale image prompts (*i.e.*, $p_\phi(c_i \mid x_{i-1}, x_{i-2})$); and (iii) prompts generated after GRPO fine-tuning (*i.e.*, $p_\phi(c_i \mid x_{i-1}, x_{i-2})$ with RL-trained $\phi$).

Example questions are provided in Fig. 9. After being given a set of instructions, each user was asked to evaluate five different sets of randomly mixed zoom sequences and five different sets of randomly mixed text generations. Users expressed their preference from 'Very Bad' to 'Very Good', and the preferences were converted to a score of 1 to 5. Resulting preference scores are shown in Fig. 10. We further conduct pair-wise t-test to confirm the statistical significance of the scores.

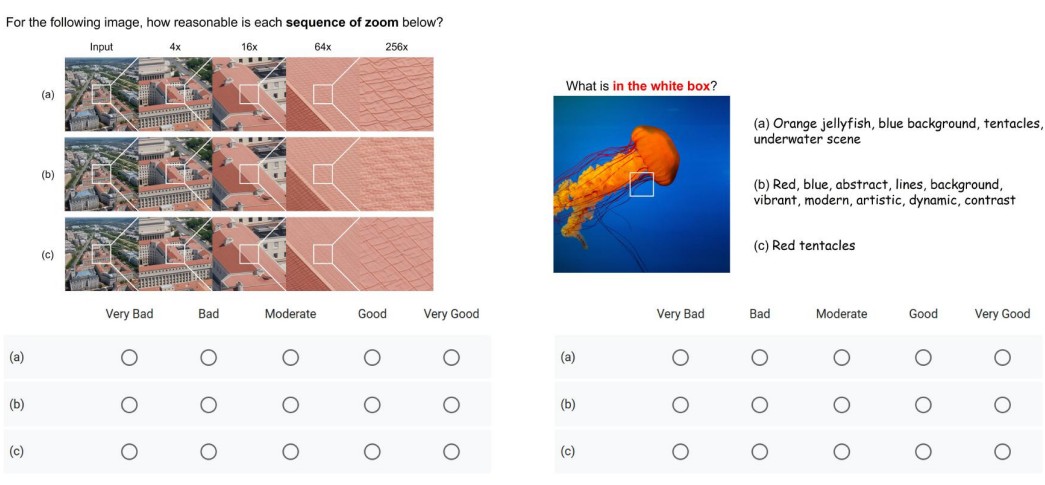

Figure 9: Example questions used for the MOS test. (Left) **Human-Preferred Image Generation.** Users were first given the instruction: 'In this survey, several samples will be given where we zoom into the center of the image. For each sequence of zoom, please rate how preferable the zoom is. (i.e., If we zoom into this input image, will the images look like this sequence?)' (Right) **Human-Preferred Text Generation.** Users were first given the instruction: 'In this section, several samples will be given where we try to explain the center of the image. For each image, please rate how preferable the explanation is. (i.e., Does the text explanation well explain what is in the white box?)'

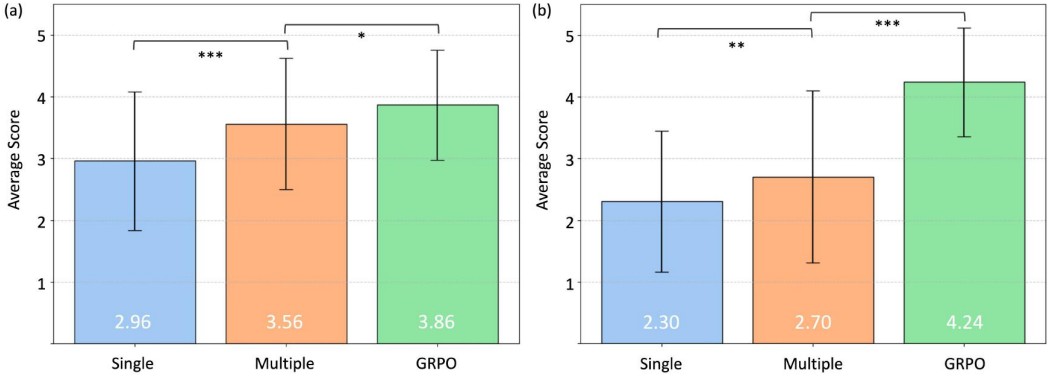

Figure 10: (a) Mean opinion scores for image generation. (b) Mean opinion scores for text generation. The scores on each bar denote the means and the error bars represent standard deviation. Significance of scores are denoted as, *: $p < 0.05$, **: $p < 0.01$, ***: $p < 0.001$.

# E   Additional Results for Performing CoZ with Open-Source OSEDiff

We further prove the applicability of our CoZ framework with the open-source OSEDiff [48] model (leveraging Stable Diffusion v2.1 backbone) available at `https://github.com/cswry/OSEDiff`. Quantitative comparison on DIV2K, DIV8K training datasets are provided in Tab. 4 and example qualitative results are provided in Fig. 11. Results show that CoZ is robust and shows good performance when utilizing OSEDiff that leverages the Stable Diffusion v2.1 model as its backbone.

Table 4: Quantitative comparison using the open-source OSEDiff. **Best**, Second-Best.

| Scale | Method | DIV2K | | | | DIV8K | | | |
|---|---|---|---|---|---|---|---|---|---|
| | | NIQE↓ | MUSIQ↑ | MANIQA↑ | CLIPIQA↑ | NIQE↓ | MUSIQ↑ | MANIQA↑ | CLIPIQA↑ |
| 4× | NN Interpolation | 12.1252 | 39.96 | 0.3396 | 0.2630 | 13.1984 | 40.26 | 0.3472 | 0.2672 |
| | Direct SR | 4.7572 | 69.26 | 0.6366 | 0.7266 | 4.8659 | 68.16 | 0.6349 | 0.7198 |
| | CoZ (Null) | 4.7295 | 69.34 | 0.6359 | 0.7272 | **4.8174** | 68.11 | 0.6332 | 0.7184 |
| | CoZ (DAPE) | 4.7577 | 69.26 | 0.6366 | 0.7265 | 4.8662 | 68.16 | **0.6350** | 0.7199 |
| | CoZ (VLM) | **4.7241** | **69.42** | **0.6368** | **0.7279** | 4.8437 | **68.31** | 0.6346 | **0.7224** |
| 16× | NN Interpolation | 22.1215 | 24.01 | 0.3378 | 0.2346 | 22.2744 | 24.94 | 0.3465 | 0.2585 |
| | Direct SR | 6.9951 | 51.88 | 0.5361 | 0.6206 | 7.4394 | 51.65 | 0.5472 | 0.6300 |
| | CoZ (Null) | 6.5369 | 61.86 | 0.5776 | **0.6988** | 6.7363 | 60.76 | 0.5842 | 0.6919 |
| | CoZ (DAPE) | 6.5628 | 61.47 | 0.5799 | 0.6899 | 6.7985 | 60.58 | 0.5888 | 0.6926 |
| | CoZ (VLM) | **6.5254** | **62.05** | **0.5801** | 0.6958 | **6.7348** | **61.11** | **0.5904** | **0.6978** |
| 64× | NN Interpolation | 27.4051 | 37.69 | 0.3803 | 0.3690 | 27.7533 | 37.13 | 0.3861 | 0.3837 |
| | Direct SR | 15.6269 | 21.56 | 0.4255 | 0.4943 | 15.8252 | 22.02 | 0.4316 | 0.5059 |
| | CoZ (Null) | 8.9369 | 54.46 | 0.5598 | **0.6672** | 8.9645 | 53.48 | 0.5643 | 0.6655 |
| | CoZ (DAPE) | 8.8681 | 53.50 | 0.5622 | 0.6553 | 9.0221 | 52.76 | 0.5687 | 0.6616 |
| | CoZ (VLM) | **8.8259** | **54.84** | **0.5645** | 0.6615 | **8.8553** | **53.84** | **0.5716** | **0.6677** |
| 256× | NN Interpolation | 34.8461 | 27.01 | 0.4179 | 0.5259 | 37.2612 | 26.98 | 0.4184 | 0.5299 |
| | Direct SR | 15.6688 | 26.37 | 0.4593 | 0.5203 | 15.9510 | 26.17 | 0.4574 | 0.5231 |
| | CoZ (Null) | 11.0907 | 47.14 | 0.5441 | 0.6223 | 11.0661 | 47.09 | 0.5439 | 0.6297 |
| | CoZ (DAPE) | 11.0014 | 45.81 | 0.5440 | 0.6162 | 10.9251 | 46.50 | 0.5475 | 0.6345 |
| | CoZ (VLM) | **10.8156** | **48.22** | **0.5495** | **0.6257** | **10.7086** | **48.25** | **0.5518** | **0.6384** |

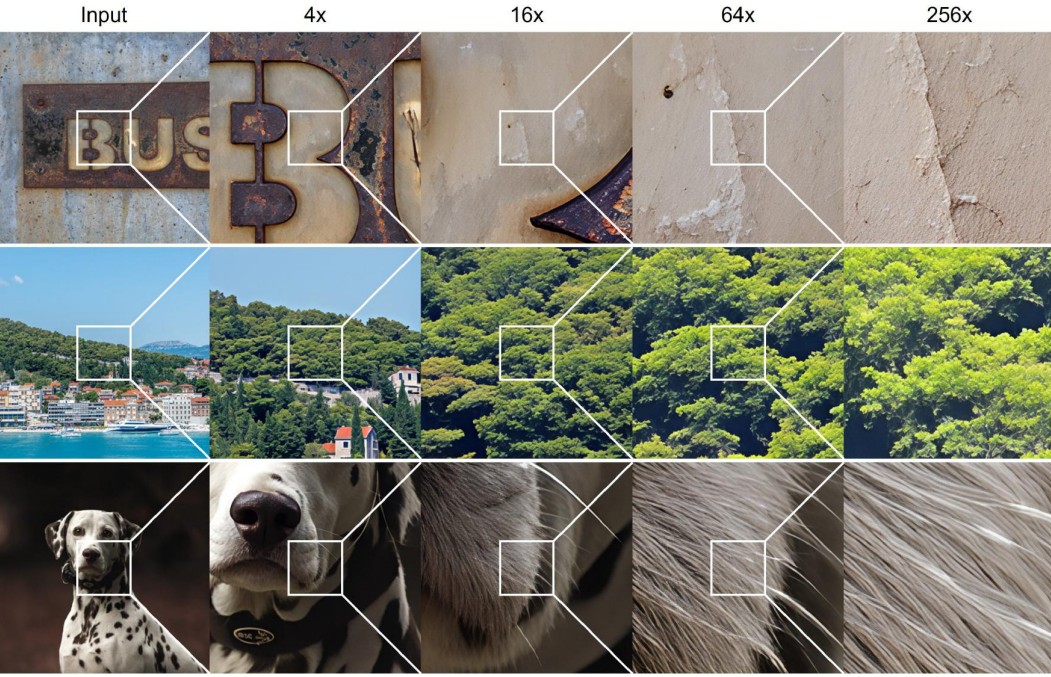

Figure 11: Qualitative results for performing CoZ with the open-source OSEDiff (leveraging Stable Diffusion v2.1 as the diffusion backbone). The GRPO fine-tuned VLM is used as the prompt extractor.

# F    Using Qwen2.5-VL-7B-Instruct as the Critic VLM

Quantitative results using Qwen2.5-VL-7B-Instruct as the critic VLM is provided in Tab. 5. All settings are identical to Tab. 1 except for choice of critic VLM.

Table 5: Quantitative results using Qwen2.5-VL-7B-Instruct as the critic VLM.

| Scale | Method | DIV2K | | | | DIV8K | | | |
|---|---|---|---|---|---|---|---|---|---|
| | | NIQE↓ | MUSIQ↑ | MANIQA↑ | CLIPIQA↑ | NIQE↓ | MUSIQ↑ | MANIQA↑ | CLIPIQA↑ |
| 4× | Direct SR | 4.7320 | **67.00** | **0.6344** | **0.7005** | 4.8631 | **66.29** | **0.6359** | **0.6946** |
| | CoZ (VLM) | **4.6546** | 66.89 | 0.6336 | 0.6993 | **4.7947** | 66.19 | 0.6349 | 0.6939 |
| 16× | Direct SR | 7.2183 | 51.25 | 0.5406 | 0.6080 | 7.5855 | 50.17 | 0.5473 | 0.6035 |
| | CoZ (VLM) | **6.2600** | **58.44** | **0.5955** | **0.6520** | **6.5513** | **57.62** | **0.5996** | **0.6572** |
| 64× | Direct SR | 16.5915 | 22.54 | 0.3995 | 0.4309 | 16.5874 | 22.97 | 0.4069 | 0.4451 |
| | CoZ (VLM) | **7.8554** | **52.12** | **0.5824** | **0.6229** | **8.0089** | **51.07** | **0.5826** | **0.6235** |
| 256× | Direct SR | 16.1749 | 28.89 | 0.4470 | 0.5196 | 15.8667 | 28.90 | 0.4464 | 0.5256 |
| | CoZ (VLM) | **8.9237** | **48.60** | **0.5786** | **0.5948** | **8.8982** | **48.06** | **0.5767** | **0.6040** |

# G    Zooming into Overlapping Regions

Cosine similarity measurements of overlapping patches across scales are provided in Tab. 6. Specifically, for the first 500 images of the DIV2K dataset, we create overlapping patch pairs with overlapping amounts of 50% (*i.e.*, the right half of one patch and the left half of the other patch represent the same region in the original image). The pixels of overlapping patches are flattened into a 1-dimensional vector and their cosine similarity are calculated. We evaluate for patch pairs across different magnifications (multiple recursions), and observe error accumulation. Cosine similarity for non-overlapping patches are also reported as reference. Results show that cosine similarity of overlapping patches are high, even for the highest magnification of $256\times$.

Table 6: Cosine similarity for overlapping/non-overlapping regions.

| Scale | Overlapping Regions | | | Non-overlapping Regions | | |
|---|---|---|---|---|---|---|
| | Mean | Median | Minimum | Mean | Median | Minimum |
| 4× | 0.9977 | 0.9982 | 0.9864 | 0.7951 | 0.8196 | 0.3077 |
| 16× | 5.5208 | 6.6210 | 0.5387 | 0.5535 | 5.5940 | 0.5346 |
| 64× | 4.9803 | 6.6210 | 0.6361 | 0.6596 | 5.0305 | 0.6328 |
| 256× | 4.8637 | 6.6210 | 0.6459 | 0.6835 | 4.9414 | 0.6405 |

# H    Full Image Super-Resolution

Chain-of-Zoom can be applied to super-resolution of full images by using the same tiling method used in prior works (*e.g.*, SeeSR, OSEDiff). Specifically, we use tiling of VAE and tiling of latents as described below to produce full-image super-resolution results without boundary artifacts.

1. A given LR image is resized to the target resolution and VAE encoding is done in tiles, allowing encoding to be performed even in settings of limited GPU memory.

2. The encoded (low-resolution) latent is tiled into overlapping patches. For latent sizes of $64 \times 64$, we find overlaps of 16 to work sufficiently well.

3. Each low-resolution patch of $64 \times 64$ passes through the super-resolution network to become high-resolution patches, each guided by patch-specific prompts generated by the prompt-extractor VLM. Note that this step requires multiple passes of the VLM, a computational bottleneck to be solved by future work.

4. The output high-resolution patches are multiplied by Gaussian weights in overlapping regions for smooth transposition between patches, and then combined to create the final high-resolution image.

5. The whole process is repeated as *scale autoregression* to achieve higher resolutions.

# I Additional Qualitative Results

Additional qualitative results of extreme super-resolution by CoZ are provided below.

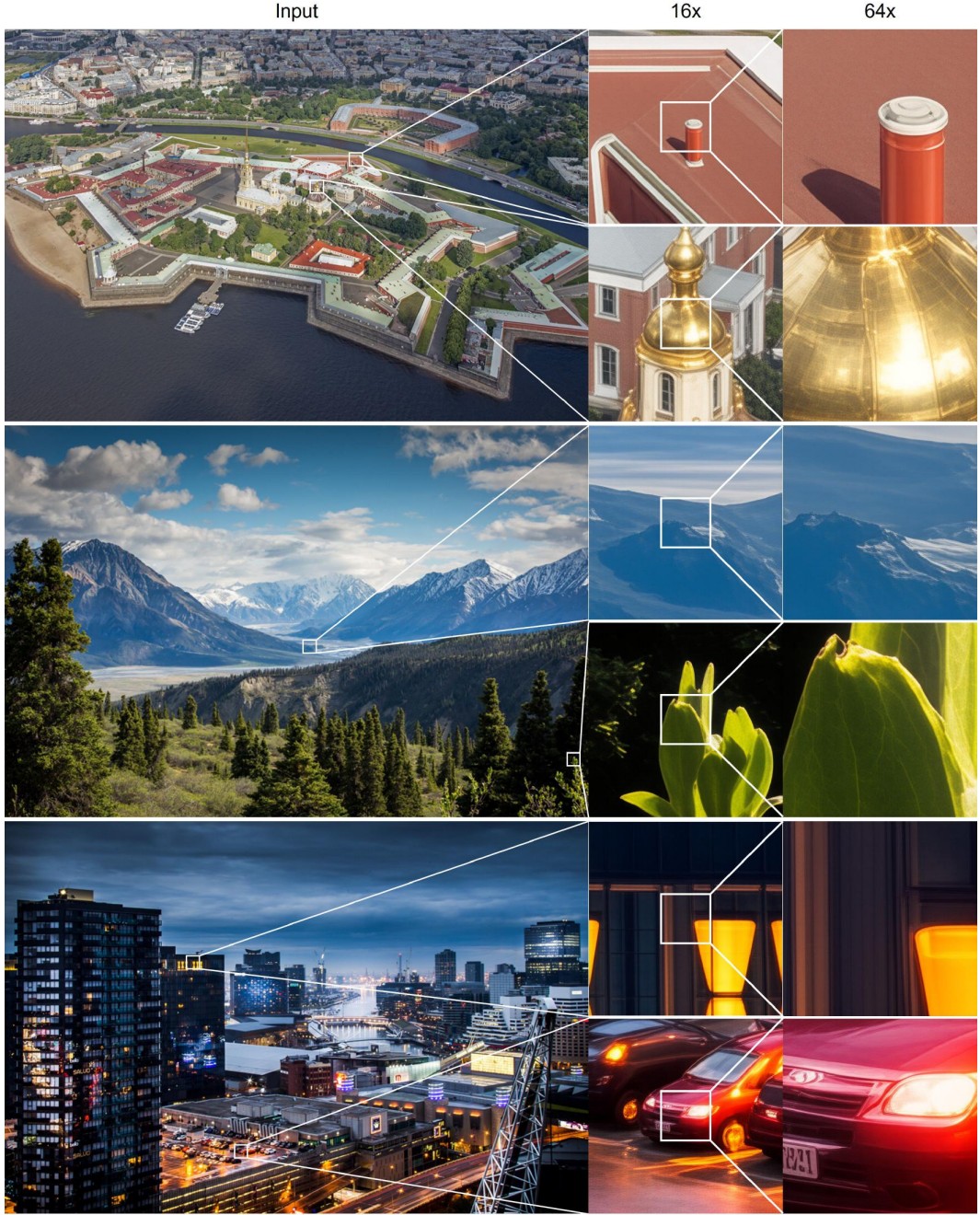

Figure 12: Extreme super-resolution of photorealistic images by CoZ up to $64\times$ magnification.

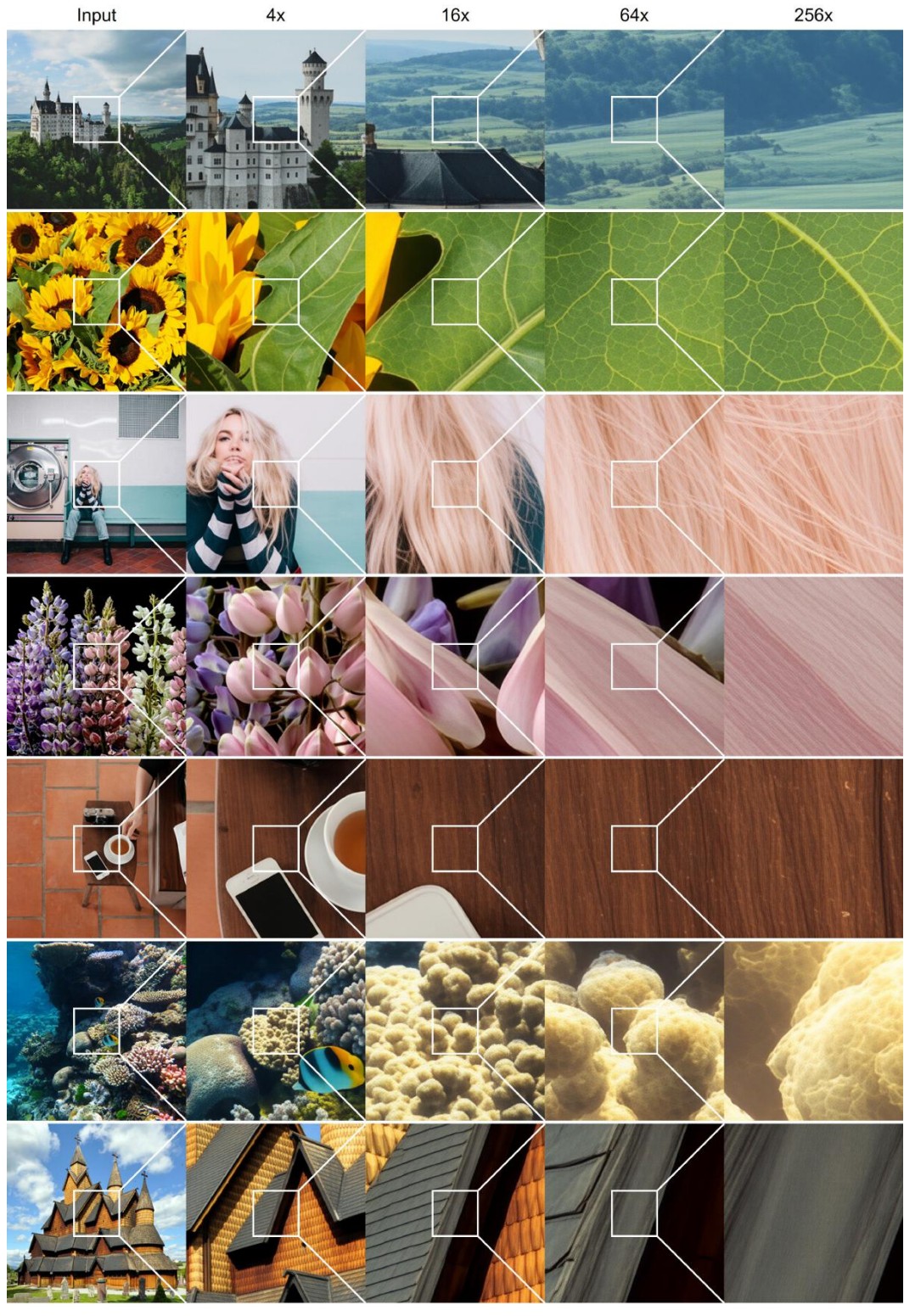

Figure 13: Extreme super-resolution of photorealistic images by CoZ up to $256\times$ magnification.

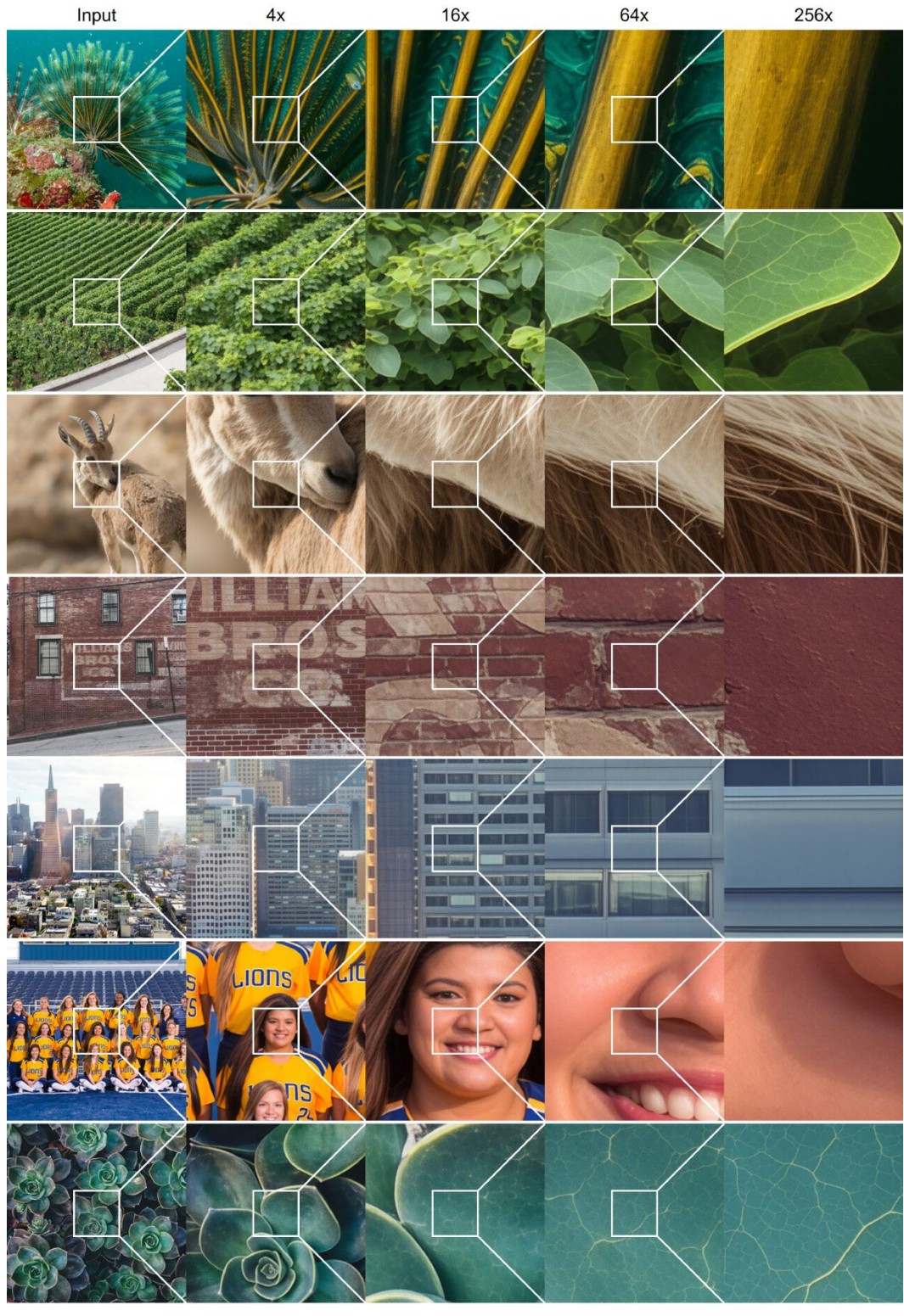

Figure 14: Extreme super-resolution of photorealistic images by CoZ up to $256\times$ magnification.

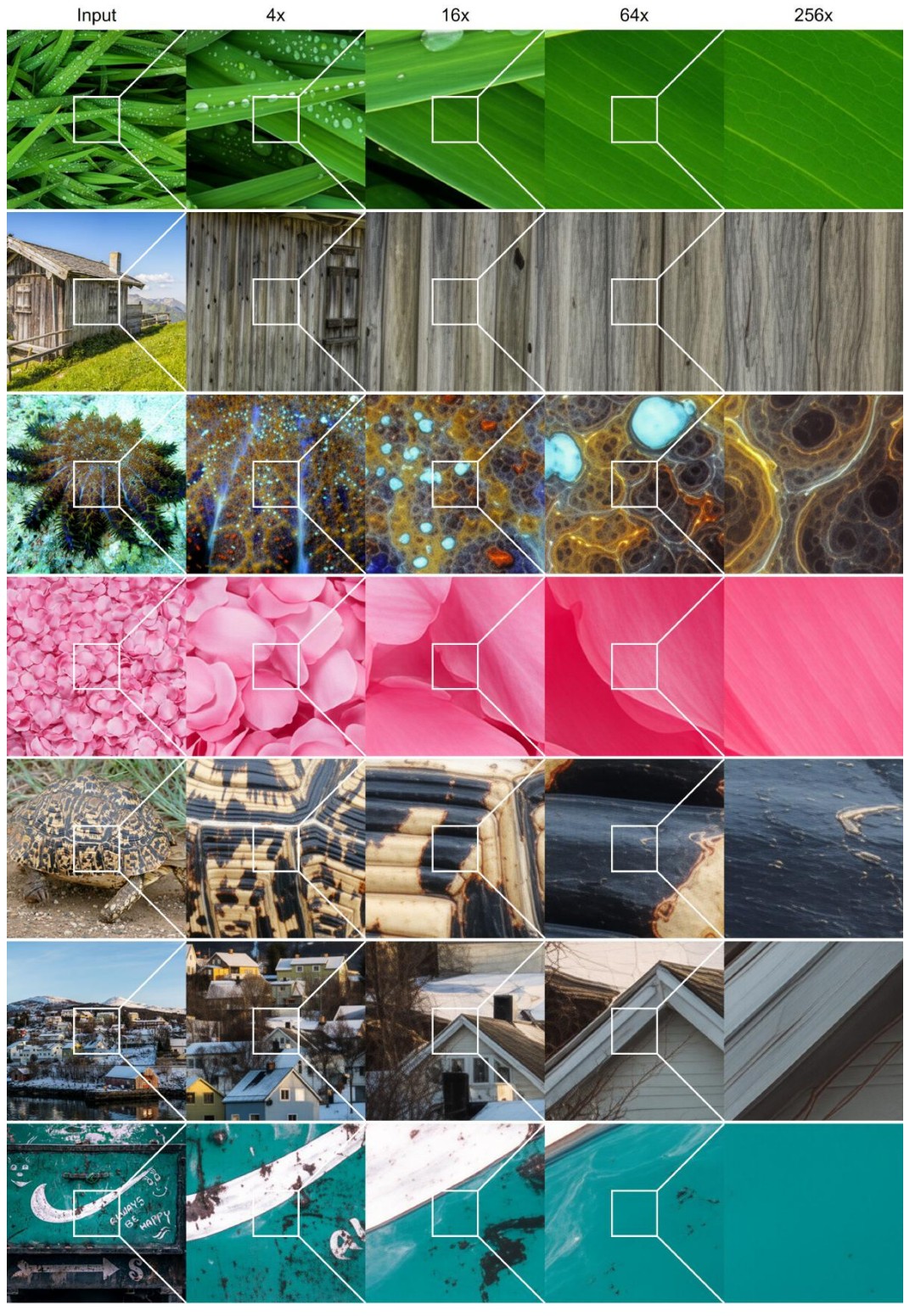

Figure 15: Extreme super-resolution of photorealistic images by CoZ up to $256\times$ magnification.

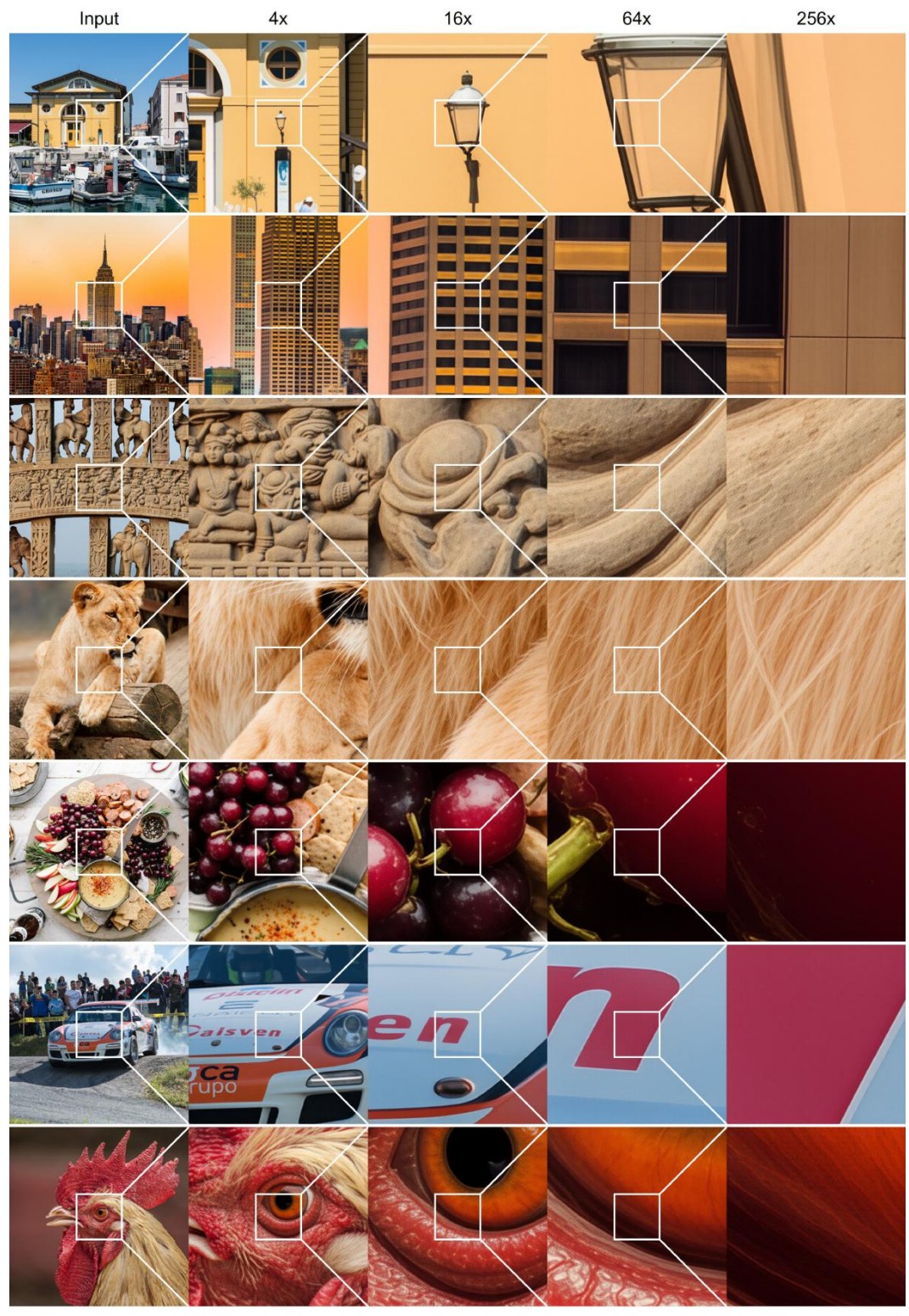

Figure 16: Extreme super-resolution of photorealistic images by CoZ up to $256\times$ magnification.

# J  Example Failure Modes of VLM before Fine-Tuning

## J.1  Repetition

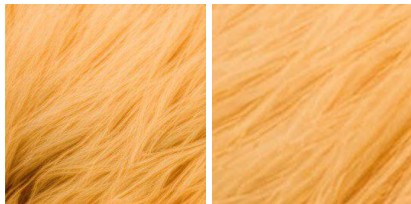

The second image is a zoom-in of the first image. Based on this knowledge, what is in the second image? Give me a set of words.

---

**RESPONSES**

**Qwen2.5-VL-3B-Instruct**  fur texture orange background animal fur close-up pattern
texture orange fur texture orange fur background orange fur
texture orange fur background orange fur texture orange fur
texture orange fur texture orange fur texture orange fur texture
orange fur texture orange fur texture orange fur texture orange
fur texture orange fur texture orange fur texture orange fur ...

**+ GRPO Training**  fur

## J.2  Unwanted Phrases

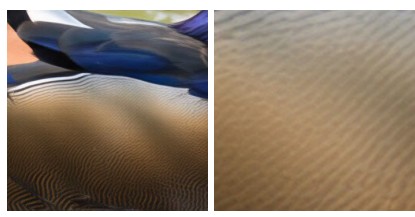

The second image is a zoom-in of the first image. Based on this knowledge, what is in the second image? Give me a set of words.

---

**RESPONSES**

**Qwen2.5-VL-3B-Instruct**  The second image shows a close-up view of a surface with a textured pattern. The texture appears to be a combination of smooth and slightly raised areas, giving it a somewhat wavy or ripple-like appearance. The color gradient ranges from a lighter shade at the top to a darker shade at the bottom, creating a sense of depth and dimension.

**+ GRPO Training**  feathers

## J.3 Suboptimal Results

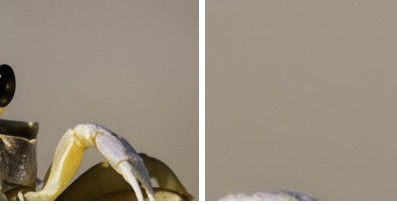

The second image is a zoom-in of the first image. Based on this knowledge, what is in the second image? Give me a set of words.

**RESPONSES**

**Qwen2.5-VL-3B-Instruct**    ant leg

**+ GRPO Training**    crab claw

