# OpenReview forum: "Chain-of-Zoom: Extreme Super-Resolution via Scale Autoregression and Preference Alignment"
_NeurIPS.cc/2025/Conference — NeurIPS 2025 spotlight_

### Official Review · Reviewer_yu7b · 2025-06-28

**Clarity:** 3
**Significance:** 2
**Originality:** 2
**Rating:** 4
**Confidence:** 4

**Summary:**

The paper introduces Chain-of-Zoom (CoZ), a framework that applies cascaded image SR on center-cropped SR reconstructions.
Instead of just taking the LR (previous zoom state), it proposes to use the last two zoom states to generate a text prompt, which makes it technically different from how SeeSR is formulated (despite being applied to another kind of task).
Also,  the VLM is fine-tuned via Generalized Reward Policy Optimization (GRPO) against a critic VLM.

**Questions:**

- How would the proposed VLM applied in SeeSR perform in classical image SR settings?
- What is the difference between single-step SR and your proposed double-step SR (1 zoom state vs 2 zoom states) quantitatively?
- If you Zoom in two patches that share overlap, how well/consistent is the overlap in the respective SR patches? How does this error accumulate over multiple Zooms?
- Could you highlight for me why this task was invented in the first place and is not compared to SeeSR, but cascaded?

While the authors discuss error accumulation and potential misuse (e.g., reconstructing sensitive imagery), they do not compare to classical prompt-aware SR baselines (e.g., SeeSR) or isolate the benefits of VLM prompts in a non-chain setting.
I suggest adding these comparisons to fully characterize CoZ’s novelty and impact.
So far, it feels like the authors invented a new task in order to justify their proposed method.
As a result, I tend towards reject but I might change my mind based on the author's reply.

**Ethical Concerns:**

["NO or VERY MINOR ethics concerns only"]

**Final Justification:**

All raised concerns were addressed accordingly and the newly added results/tables further improved the quality of the submission (see Q1-Q3)! I especially appreciate the provided experiments regarding cosine similarity!

**Limitations:**

yes

**Paper Formatting Concerns:**

All fine

**Quality:**

3

**Strengths And Weaknesses:**

Strengths:
- Solid probabilistic formulation (AR-2 factorization).
- comprehensive evaluation using no-reference perceptual metrics (NIQE, MUSIQ, MANIQA-pipal, CLIPIQA)

Weakness:
- No comparison to the state-of-the-art by reformulating the problem of cascaded image SR on center-cropped SR reconstructions.

---

> ### Author Rebuttal · Authors · 2025-07-31
>
> **Q4. [yu7b] questions the motivation of the work, and the reason for its cascaded approach without being compared to SeeSR.**
>
> We would like to start by answering the last question the reviewer raised, so as to help give the reviewer a better understanding of the motivation of our work before delving into the other concerns raised. The motivation behind our work stems from a fundamental limitation in existing single-image super-resolution (SISR) models, such as SeeSR: **they are inherently constrained by their training configurations, and typically fail or produce severe artifacts when pushed beyond these scales.** To support this claim, we provide the quantitative results of attempting super-resolution over increasing magnifications with SeeSR in the response below (please refer to the response for W1 below).
>
> Although SeeSR shows strong performance at low 4x magnification, performance is degraded at magnifications higher than its training regime (*e.g.*, 16x, 64x, 256x). In this work, our motivation is to solve this problem and effectively utilize pretrained SR models to achieve much higher magnifications than they are originally trained for. Thus, we did not originally compare with SeeSR as our method is focused on *going **beyond** the training regime* of SR models, rather than *achieving superior performance **within** the training regime* for super-resolution. It is also worth noting that directly training models for extreme magnification factors (*e.g.*, 64× or 256×) is computationally prohibitive due to immense resource requirements, making such approaches practically infeasible.
>
> The main objective of our work, therefore, is to overcome this scalability bottleneck by introducing Chain-of-Zoom (CoZ), a framework designed specifically to enable existing, well-trained SR models (like SeeSR) to achieve significantly higher resolutions through a autoregressive process. CoZ accomplishes this by breaking down extreme SR tasks into manageable intermediate stages, each guided by semantically coherent multi-scale prompts. By design, **CoZ complements rather than competes with models like SeeSR**; it effectively broadens their capabilities to achieve magnifications beyond their original operational range without additional training. Hence, CoZ is not intended as a replacement but as a versatile and resource-efficient framework that substantially extends the practical applicability and scalability of existing super-resolution models.
>
> In summary,  our **multi-scale-aware, GRPO fine-tuned VLM** is introduced as a novel method to *aid* the aforementioned autoregressive process. Based on the observation that text prompts are important for creating high-frequency details, we leverage the novel prompt-extraction VLM as a *replacement* for DAPE (a widely used prompt-extractor that can be found in SeeSR, OSEDiff, etc.) and prove its effectiveness through comprehensive experiments.
>
> Below, we further provide detailed point-by-point responses to the other concerns raised.
>
> ---
> **W1. [yu7b] suggests adding comparisons with state-of-the-art SR baselines.**
>
> We would like to remind the reviewer that quantitative and qualitative results using the another (official) implementation of **OSEDiff** (with Stable Diffusion v2.1 as the backbone) is provided in **Section E of Supplementary Material**. We show that applying CoZ greatly improves performance of OSEDiff, a state-of-the-art SR method, especially at high magnifications.
>
> Below, we further provide **quantitative results for other baseline methods**: arbitrary-scale SR methods (**LIIF [1]**) and diffusion-based SR methods (**SeeSR, S3Diff [2]**). The quantitative results for SeeSR was provided in the previous response, and results for other baselines are given below. All three baselines show greatly degraded performance at high magnifications. For the case of S3Diff, we additionally show **quantitative results for applying CoZ to the pretrained, freely accessible S3Diff, leveraging our multi-scale aware GRPO fine-tuned VLM**. All the results clearly confirm the significant performance improvement by CoZ.
>
> Quantitative results for **LIIF**, **SeeSR**, and **S3Diff**:
> |||DIV2K||||DIV8K||||
> |---|---|---|---|---|---|---|---|---|---|
> |Scale|Method|NIQE ↓|MUSIQ ↑|MANIQA ↑|CLIPIQA ↑|NIQE ↓|MUSIQ ↑|MANIQA ↑|CLIPIQA ↑|
> |4x|LIIF|6.6210 |57.47|0.5456|0.4587|6.8050 |55.52|0.5386|0.4545|
> |4x|SeeSR|5.5208 |57.56|0.5387|0.5535|5.5940 |55.50|0.5346|0.5385|
> |4x|S3Diff|4.9803 |65.82|0.6361|0.6596|5.0305 |64.08|0.6328|0.6481|
> |4x|S3Diff + Ours|**4.8637**|**67.18**|**0.6459**|**0.6835**|**4.9414**|**65.31**|**0.6405**|**0.6680**|
> |16x|LIIF|11.4815|25.94|0.2860|0.3024|11.8734|26.73|0.2896|0.3178|
> |16x|SeeSR|9.6798 |41.68|0.4310|0.4669|9.8865 |40.02|0.4210|0.4601|
> |16x|S3Diff|6.7383 |51.14|0.5305|0.5886|7.2399 |50.32|0.5370|0.5841|
> |16x|S3Diff + Ours|**6.7310**|**56.82**|**0.5752**|**0.6168**|**6.8698**|**56.46**|**0.5836**|**0.6218**|
> |64x|LIIF|16.9215|20.02|0.3451|0.4131|17.4912|20.43|0.3522|0.4250|
> |64x|SeeSR|16.9095|21.83|0.4106|0.4193|16.9275|22.68|0.4167|0.4309|
> |64x|S3Diff|16.1421|21.54|0.3893|0.4917|16.5506|22.00|0.3968|0.5089|
> |64x|S3Diff + Ours|**8.7770**|**48.90**|**0.5490**|**0.5801**|**8.9794**|**48.07**|**0.5560**|**0.5909**|
> |256x|LIIF|23.8949|26.05|0.4108|0.5380|24.3300|26.01|0.4116|0.5423|
> |256x|SeeSR|20.9635|25.81|0.4438|0.5193|19.9628|25.94|0.4429|0.5203|
> |256x|S3Diff|16.7809|25.92|0.4324|0.5359|16.9952|25.91|0.4312|0.5415|
> |256x|S3Diff + Ours|**10.7668**|**43.59**|**0.5438**|**0.5570**|**10.5001**|**43.31**|**0.5417**|**0.5673**|
>
> ---
> **Q1. [yu7b] questions about applying the proposed VLM to SeeSR.**
>
> The direct integration of our proposed VLM into SeeSR requires additional complexity due to the architectural requirements of SeeSR, specifically its reliance on ControlNet architecture. This confirms the efficiency of our CoZ scheme.  Specifically, SeeSR necessitates both "soft" and "hard" embeddings: the "soft embeddings" (continuous embeddings) can only be generated by the Degradation-Aware Prompt Extractor (DAPE). Since the proposed VLM generates discrete textual prompts (which correspond only to "hard embeddings"), simply replacing DAPE with the proposed VLM within the SeeSR architecture is not feasible without significant architectural modifications.  Furthermore, we also delivered proper comparison between DAPE and our proposed VLM in **Figure 5** and **Table 1** of the main paper, confirming the performance improvements of our proposed VLM.
>
> ---
> **Q2. [yu7b] questions the quantitative difference between single-step SR and double-step SR.**
>
> The reviewer is kindly reminded  that the quantitative results in **Table 1** of the main paper already contained the difference between single-step SR and double-step SR. Specifically, 'Direct SR' represents reaching the target magnification (*i.e.*, one of 4x, 16x, 64x, 256x) in a single step, and quantitative results show how image quality becomes degraded at high magnifications. Meanwhile 'CoZ' represents our proposed method, in which we do not attempt to reach the target magnification in a single pass, and split the SR process into smaller (easier) steps. While difference seems small at lower magnifications, the quantitative difference between 'Direct SR' and 'CoZ' become significant as we reach higher magnifications.
>
> Thus, in answer to the question of "quantitative difference between single-step SR and double-step SR", we could say that the 16x magnification scenario specifically represents this case as we decompose the target 16x magnification into two steps of 4x magnification. Compared to attempting 16x magnification in one step (as denoted by 'Direct SR'), we show that using the Chain-of-Zoom framework produces superior results. The comparison table of Quantitative results for LIIF, SeeSR, and S3Diff above also confirmed the significant performance gain by CoZ.
>
> ---
> **Q3. [yu7b] inquires of results for zooming into two patches that show overlap, especially across multiple zooms.**
>
> We thank the reviewer for the intriguing suggestion, and provide **cosine similarity results** for overlapping patches of 50% across scales. Specifically, for the first 500 images of the DIV2K dataset, we create overlapping patch pairs with overlapping amounts of 50% (*i.e.*, the right half of one patch and the left half of the other patch represent the same region in the original image). The pixels of overlapping patches are flattened into a 1-dimensional vector before cosine similarity calculation. We evaluate for patch pairs across different magnifications (multiple recursions), and observe error accumulation. Furthermore, we also report cosine similarity for non-overlapping patches as reference. Results show that cosine similarity of overlapping patches are high, even for the highest magnification of 256x. This confirms the semantically consistent guiding capability of CoZ.
>
> Cosine similarity results for overlapping regions:
> |Scale|Cosine Similarity (Mean)|Cosine Similarity (Median)|Cosine Similarity (Minimum)|
> |---|---|---|---|
> |4x|0.9977|0.9982|0.9864|
> |16x|0.9962|0.9976|0.9546|
> |64x|0.9925|0.9966|0.9048|
> |256x|0.9484|0.9829|0.2408|
>
> Cosine similarity results for non-overlapping regions:
> |Scale|Cosine Similarity (Mean)|Cosine Similarity (Median)|Cosine Similarity (Minimum)|
> |---|---|---|---|
> |4x|0.7951|0.8196|0.3077|
> |16x|0.8279|0.8613|0.3003|
> |64x|0.8722|0.9306|0.1082|
> |256x|0.9121|0.9665|0.2349|
>
> ---
> **References**
>
> [1] Learning Continuous Image Representation with Local Implicit Image Function, CVPR 2021
>
> [2] Degradation-Guided One-Step Image Super-Resolution with Diffusion Priors

---

> > ### Comment · Reviewer_yu7b · 2025-08-03
> >
> > Dear authors,
> >
> > thank you for your detailed explanations! I hope these new findings can find their way into the final manuscript, since it became a much stronger submission with them. Like promised, I'm happy to increase my final rating as the authors addressed and explained every raised concern!

---

> > > ### Author Response · Authors · 2025-08-03
> > > **Thanks for increasing final rating!**
> > >
> > > Thank you very much for your positive response and for deciding to raise the final rating. We're pleased to hear that  our explanation has helped make the paper much stronger. If you have any further questions or comments, we would be happy to discuss them.

---

### Official Review · Reviewer_yTPV · 2025-06-29

**Clarity:** 3
**Significance:** 3
**Originality:** 3
**Rating:** 5
**Confidence:** 4

**Summary:**

This work proposed a new autoregressive framework, CoZ, for extreme-scale SISR tasks. CoZ is motivated by the observation that most SR models are scale-specific and struggle to generalize to unseen, extreme magnification levels. To address that, CoZ decomposes the LR to HR transformation into a sequence of intermediate scale-states and multi-scale-aware prompts. These prompts are generated by a VLM, which is fine-tuned using a GRPO-based RL pipeline to align with human preferences. Experiment results show that CoZ can effectively reconstruct sharp and realistic HR images from LR inputs, even at extreme scales.

**Questions:**

1. Since there is no ground truth for high-scale results (e.g., 64×, 256×), how do you know the model is not hallucinating unrealistic details？The results in Figures 3 and 5 look good, but it's hard to know how far they are from the real high-res images.

2. Have you measured the computational overhead of the full CoZ pipeline (especially at high scales like 256×)? How does inference time or memory consumption compare with a standard one-step SR method?

3. Why is the AR-2 modeling clearly better than a simpler Markov setup?
In lines 128–130, you state that "we **find** that multi-scale-aware text extraction is necessary by feeding \(x_{i-1}\) and the coarser state \(x_{i-2}\) into prompt generation."  Do you have any theoretical justification or quantitative evidence to support this design choice?

**Ethical Concerns:**

["NO or VERY MINOR ethics concerns only"]

**Final Justification:**

The authors addressed my main concerns clearly. The system is less complex than they expected, hallucination is mitigated via prompt alignment with human preferences, and the computational overhead is justified. AR-2 modeling is supported by metrics and user studies. The novelty and results are sufficient to acceptance.

**Limitations:**

yes

**Quality:**

3

**Strengths And Weaknesses:**

Strengths

1.Motivation is clear and practical: they try to tackle the limitation of most SR models: work only at fixed scales and perform poorly when asked to upscale far beyond what they were trained on. Super useful and interesting.

2.Well-written and organized: the paper is easy to follow, with clear explanations and well-structured sections.

3.Promising results: the results show that the CoZ can produce sharp and natural-looking images, even when zooming in by 64× or more

Weaknesses:

1.The pipeline is complex and expensive to train: although the authors claim the method doesn’t require retraining the SR model, the full pipeline includes: a base SR model (like OSEDiff), a VLM for generating prompts, a separate critic VLM for training the prompt model with reinforcement learning. Plus, the model needs to run multiple inference steps for high magnifications. All of this adds up to high training cost and system complexity.

2.The experimental section feels somewhat limited. In particular, it's difficult to assess whether the model suffers from hallucination issues based on the qualitative results alone, since there is no ground-truth reference available for comparison at extreme scales(e.g. figure 5 and 3.).

---

> ### Author Rebuttal · Authors · 2025-07-31
>
> **W1. [yTPV] comments about the pipeline requiring high training cost and being complex.**
>
> We clarify that our Chain-of-Zoom is a method to boost the performance of pretrained SR models when they are asked to solve for resolutions beyond their training regime. Accordingly, it is a framework designed to **wrap around existing SR models** without need for retraining. Thus, the only training target is a prompt-extraction VLM, chosen to be Qwen2.5-VL-3B-Instruct in this work. We point out that the chosen Qwen2.5-VL-3B-Instruct model is relatively quite small in the domain of VLMs, and we further reduce training costs by performing LoRA fine-tuning instead of full fine-tuning. We further mention in **Section B of Supplementary Material** that we use four NVIDIA GeForce RTX 3090 GPUs for the GRPO fine-tuning, and we believe this to be relatively affordable compared to similar baselines (*e.g.*, DAPE is trained with eight NVIDIA Tesla 32G-V100 GPUs).
>
> Regarding system complexity, we would like to point out that the SR model is *not* used when fine-tuning the VLM, and that the critic VLM is *not* used during inference time. This simplifies system complexity, ensuring that **no more than two distinct models are used during either training or inference**.
>
> Furthermore, once the VLM is fine-tuned, we can utilize it in the CoZ inference framework for *any* SR backbone model. The final fine-tuned VLM can be used in a **plug-and-play manner** in combination with any type of SR model, and does not require any retraining afterwards. We prove this by showing its effectiveness when combined with various types of SR models. Apart from our main results using a custom-trained SR model, quantitative and qualitative results using the official implementation of **OSEDiff** (with Stable Diffusion v2.1 as the backbone) is provided in **Section E of Supplementary Material**. We further provide quantitative results for applying CoZ to the pretrained, official implementation of **S3Diff [1]**, leveraging our **multi-scale aware GRPO fine-tuned VLM** in the table below.
>
> Quantitative results for **S3Diff**:
> |||DIV2K||||DIV8K||||
> |---|---|---|---|---|---|---|---|---|---|
> |Scale|Method|NIQE↓|MUSIQ↑|MANIQA↑|CLIPIQA↑|NIQE↓|MUSIQ↑|MANIQA↑|CLIPIQA↑|
> |4x|S3Diff|4.9803|65.82|0.6361|0.6596|5.0305 |64.08|0.6328|0.6481|
> |4x|S3Diff + Ours|**4.8637**|**67.18**|**0.6459**|**0.6835**|**4.9414**|**65.31**|**0.6405**|**0.6680**|
> |16x|S3Diff|6.7383|51.14|0.5305|0.5886|7.2399 |50.32|0.5370|0.5841|
> |16x|S3Diff + Ours|**6.7310**|**56.82**|**0.5752**|**0.6168**|**6.8698**|**56.46**|**0.5836**|**0.6218**|
> |64x|S3Diff|16.1421|21.54|0.3893|0.4917|16.5506|22.00|0.3968|0.5089|
> |64x|S3Diff + Ours|**8.7770**|**48.90**|**0.5490**|**0.5801**|**8.9794**|**48.07**|**0.5560**|**0.5909**|
> |256x|S3Diff|16.7809|25.92|0.4324|0.5359|16.9952|25.91|0.4312|0.5415|
> |256x|S3Diff + Ours|**10.7668**|**43.59**|**0.5438**|**0.5570**|**10.5001**|**43.31**|**0.5417**|**0.5673**|
>
> ---
> **W2, Q1. [yTPV] inquires about experimental results, especially regarding the lack of ground-truth references available for comparison at extreme scales.**
>
> We appreciate the reviewer's valuable feedback regarding concerns about potential hallucination issues at extreme super-resolution scales. Indeed, by its nature, super-resolution (SR) involves synthesizing plausible high-frequency details based on learned real-world data priors. Therefore, all SR methods inherently exhibit some degree of "hallucination." However, what distinguishes effective SR methods, including ours, is their ability to generate **realistic and semantically coherent details**.
>
> The critical factor here is **semantic consistency and controllability**. Our method explicitly addresses this by integrating **multi-scale-aware text prompts**, which guide the synthesis of details at each step of the resolution enhancement process. This design ensures that while the model does hallucinate high-frequency details, these details remain semantically controlled and coherent with the context provided by VLM-generated prompts. This semantic control significantly mitigates arbitrary or inconsistent hallucinations and provides interpretable and consistent SR outcomes even at extreme magnifications.
>
> In particular, we use **human preference** as a target for alignment and fine-tune the prompt extraction VLM with GRPO, using the scores from a critic VLM as a proxy of human preference. Aside from our qualitative results, the **MOS (mean-opinion-score) test** result available in **Section D of Supplementary Material** is an important quantitative result commonly used in perceptual SR literature (FAN [2], SRGAN [3], PULSE). Results support that our method indeed produces realistic details through human preference alignment.
>
> Furthermore, in the following Tables we include **additional quantitative results** to assess the quality of prompts created by different VLMs while accounting for human preference. Specifically, we include quantitative results for the metrics **ImageReward** [4] and **HPSv2** [5] to assess the quality of VLM prompts used for zooming into images at extreme magnifications. We compare the alignment between the generated prompt and the original input image, to accurately assess the preservation of semantic consistency.
>
> Results for 64x magnification:
>
> ||DIV2K||DIV8K||
> |---|---|---|---|---|
> ||ImageReward &uarr;|HPSv2 &uarr;|ImageReward &uarr;|HPSv2 &uarr;|
> |Single|-1.9009|0.1784|-1.7699|0.1811|
> |Multi|-1.5222|0.1978|-1.3238|0.2033|
> |GRPO|-1.3558|**0.2087**|-1.2825|**0.2099**|
> |Critic|**-1.2480**|0.1900|**-1.1219**|0.1931|
>
> Results for 256x magnification:
>
> ||DIV2K||DIV8K||
> |---|---|---|---|---|
> ||ImageReward &uarr;|HPSv2 &uarr;|ImageReward &uarr;|HPSv2 &uarr;|
> |Single|-1.9705|0.1753|-1.8706|0.1777|
> |Multi|-1.7624|0.1850|-1.6635|0.1874|
> |GRPO|-1.6336|**0.2045**|-1.5634|**0.2041**|
> |Critic|**-1.4764**|0.1781|**-1.3353**|0.1816|
>
> For each row: **Single** refers to using prompts generated from only the LR input, i.e. $p_\phi(c_i|x_{i-1})$; **Multi** refers to using prompts generated from multi-scale image prompts, i.e. $p_\phi(c_i|x_{i-1},x_{i-2})$; **GRPO** refers to using $p_\phi(c_i|x_{i-1},x_{i-2})$ with GRPO fine-tuned $\phi$; **Critic** refers to using multi-scale image prompts generated by the critic VLM. Results support that GRPO fine-tuning of our prompt-extraction VLM helps human preference alignment, thus reducing any unwanted hallucinations.
>
> ---
> **Q2. [yTPV] suggests measuring the computational overhead of the full CoZ pipeline, in comparison with a standard one-step SR method.**
>
> We are grateful for the constructive feedback, and we give detailed runtime analysis for each method across varying scales (4x ~ 256x) in the table below. The average inference time required (in seconds) to apply CoZ on a single image is evaluated on 500 images of the DIV2K dataset. We divide inference into two phases: super-resolution (SR) and prompt extraction (PE); accurate analysis on the computational time required for each phase is given accordingly. Regarding memory consumption, loading and running the VLM model during inference time requires around 8GB of additional memory.
>
> |Scale|Phase|OneStep (DAPE)|CoZ (Null)|CoZ (DAPE)|CoZ (VLM)|
> |---|---|---|---|---|---|
> |4x|SR|0.1467|0.1443|0.1460|0.1445|
> |4x|PE|0.0136|0.0000|0.0130|0.3777|
> |16x|SR|0.1462|0.2886|0.2912|0.2896|
> |16x|PE|0.0130|0.0000|0.0254|0.7068|
> |64x|SR|0.1462|0.4329|0.4363|0.4349|
> |64x|PE|0.0131|0.0000|0.0382|1.0301|
> |256x|SR|0.1462|0.5774|0.5816|0.5802|
> |256x|PE|0.0132|0.0000|0.0509|1.3505|
>
> ---
> **Q3. [yTPV] requests further evidence to support the design choice of AR-2 modeling compared to the Markov setup.**
>
> We would like to point out that the quantitative results regarding **semantic consistency and controllability** also serve as evidence of the importance of AR-2 modeling. For the tables below, **Single** refers to using prompts generated from only the LR input (*i.e.*, $p_\phi(c_i|x_{i-1})$), which represents the Markov setup. **Multi** refers to using prompts generated from multi-scale image prompts (*i.e.*, $p_\phi(c_i|x_{i-1},x_{i-2})$), which represents AR-2 modeling. Using AR-2 modeling improves semantic consistency of the SR process.
>
> Results for 64x magnification:
>
> ||DIV2K||DIV8K||
> |---|---|---|---|---|
> ||ImageReward &uarr;|HPSv2 &uarr;|ImageReward &uarr;|HPSv2 &uarr;|
> |Single|-1.9009|0.1784|-1.7699|0.1811|
> |Multi|**-1.5222**|**0.1978**|**-1.3238**|**0.2033**|
>
> Results for 256x magnification:
>
> ||DIV2K||DIV8K||
> |---|---|---|---|---|
> ||ImageReward &uarr;|HPSv2 &uarr;|ImageReward &uarr;|HPSv2 &uarr;|
> |Single|-1.9705|0.1753|-1.8706|0.1777|
> |Multi|**-1.7624**|**0.1850**|**-1.6635**|**0.1874**|
>
> We further mention that the user study in **Section D of Supplementary Material** also compares between the Markov setup of $p_\phi(c_i|x_{i-1})$ and AR-2 modeling of $p_\phi(c_i|x_{i-1},x_{i-2})$. For both human-preferred image generation and human-preferred text generation, the AR-2 modeling proves more effective.
>
> ---
> **References**
>
> [1] Degradation-Guided One-Step Image Super-Resolution with Diffusion Priors
>
> [2] Progressive Face Super-Resolution via Attention to Facial Landmark, BMVC 2019
>
> [3] Photo-Realistic Single Image Super-Resolution Using a Generative Adversarial Network, CVPR 2017
>
> [4] ImageReward: Learning and Evaluating Human Preferences for Text-to-Image Generation, NeurIPS 2023
>
> [5] Human Preference Score v2: A Solid Benchmark for Evaluating Human Preferences of Text-to-Image Synthesis

---

> > ### Comment · Reviewer_yTPV · 2025-08-06
> > **Official Comment by Reviewer yTPV**
> >
> > Dear authors,
> >
> > Thank you for your detailed and well-structured responses. I appreciate the clarifications and additional quantitative results provided. The rebuttal addresses my main concerns effectively, so I am updating my score from 4 to 5. I think the novelty and experiments sufficiently support acceptance to NeurIPS.

---

> > > ### Author Response · Authors · 2025-08-06
> > > **Thank you for increasing your score!**
> > >
> > > Thank you for your encouraging feedback and for increasing the final rating. We are pleased to hear that your concerns have been properly addressed. Should any other questions or comments come up, we would be glad to discuss them.

---

### Official Review · Reviewer_wycu · 2025-07-02

**Clarity:** 3
**Significance:** 3
**Originality:** 3
**Rating:** 5
**Confidence:** 4

**Summary:**

The paper addresses extreme image super-resolution (SR), with magnification levels reaching up to 256x. It introduces Chain-of-Zoom (CoZ), which decomposes extreme SR into sequential 4x SR steps using the same backbone SR model. To address the diminishing visual cues at high SR ratios, the method incorporates multi-scale-aware text prompts, where a vision-language model (VLM) generates captions for the current low-resolution (LR) image based also on its previous output. Additionally, GRPO is employed to enhance the captioning ability of the VLM in this extreme SR setting. Experiments on DIV2K and DIV8K demonstrate that the proposed CoZ method outperforms the direct SR strategy and other captioning baselines.

**Questions:**

- The paper should include time efficiency measurements for the proposed method to provide insights into its computational performance.
- The distinction between *random degradation (same settings as OSEDiff)* and *4x specific upscaling* needs to be clarified, where OSEDiff should also use 4x upscaling for training.
- The rationale for using InternVL2.5-8B as the critic VLM, which differs from the prompt-extraction VLM, should be explained. It is better the paper also explores the effects of using different critic VLMs. For example, what would the results be if Qwen2.5-VL-7B-Instruct were used as the critic VLM?
- In Figure 7, the captions for the first row and the middle row differ. This difference should be clarified.
- The paper could benefit from a discussion of related works that explore the use of text prompts to enhance SR performance.
- For Figure 5, it would be helpful to explicitly indicate the SR ratio.

**Ethical Concerns:**

["NO or VERY MINOR ethics concerns only"]

**Final Justification:**

The paper proposes a novel Chain-of-Zoom (CoZ) formulation, which decomposes extreme super-resolution (SR) into sequential 4x SR steps using the same backbone SR model. The overall quality and presentation of the paper are good. Initially, I had some questions about the experiments, but the authors' response has addressed most of my concerns. Therefore, I tend to increase my rating to Accept.

**Limitations:**

Yes.

**Quality:**

3

**Strengths And Weaknesses:**

Strengths
- The paper focuses on an interesting topic: extreme super-resolution, with magnification levels reaching up to 256x.
- It effectively decomposes extreme SR into sequential lower-ratio SR steps by utilizing the same SR backbone model. To tackle the issue of diminishing visual cues, the method introduces multi-scale-aware text prompts to provide additional semantic guidance.
- The VLM generates captions based on the current image and its previous-stage SR result, which has a larger receptive field. This process is further enhanced by GRPO, improving the SR results.
- The paper is well-structured and includes visually appealing illustrations that enhance clarity and presentation.

Weaknesses
- The validation is limited to comparisons among baselines (i.e., nearest neighbor interpolation, Direct SR, and CoZ variants: Null, DAPE, and VLM). It remains unclear how existing methods perform under similar conditions. Quantitative comparisons with existing approaches, such as diffusion-based SR methods or arbitrary-scale image super-resolution methods (and more visual comparisons), are recommended to strengthen the evaluation.
- Model training details are insufficiently explained. Specifically, it is unclear whether the SR model is jointly trained with the VLM. Additionally, the source of the caption $c$ for SR model during training (e.g., from DAPE, VLM, or fine-tuned VLM) needs to be clarified.
- For the ablation study on VLM training, the paper should provide quantitative results of different VLMs beyond the visual comparisons shown in Figure 7. Furthermore, results using a critic VLM for captions would provide a useful reference.
- The quantitative improvement (Table 1) over the baseline CoZ (Null) is relatively small.

---

> ### Author Rebuttal · Authors · 2025-07-31
>
> **W1. [wycu] recommends additional comparison with existing methods.**
>
> We would first like to remind the reviewer that quantitative and qualitative results using the official implementation of **OSEDiff** (with Stable Diffusion v2.1 as backbone) are already provided in **Section E of Supplementary Material**. Applying CoZ greatly improves performance of OSEDiff, a SOTA diffusion-based SR method.
>
> We further provide **quantitative results for other baseline methods**: arbitrary-scale SR methods (**LIIF [1]**) and diffusion-based SR methods (**SeeSR, S3Diff [2]**). All baselines show greatly degraded performance at high magnifications. We additionally show quantitative results for **applying CoZ to the pretrained S3Diff**, leveraging our multi-scale aware GRPO fine-tuned VLM. We will also include visual comparisons against baselines in the final manuscript as recommended.
>
> Table of Quantitative results for **LIIF**, **SeeSR**, and **S3Diff**:
> |||DIV2K||||DIV8K||||
> |---|---|---|---|---|---|---|---|---|---|
> |Scale|Method|NIQE ↓|MUSIQ ↑|MANIQA ↑|CLIPIQA ↑|NIQE ↓|MUSIQ ↑|MANIQA ↑|CLIPIQA ↑|
> |4x|LIIF|6.6210 |57.47|0.5456|0.4587|6.8050 |55.52|0.5386|0.4545|
> |4x|SeeSR|5.5208 |57.56|0.5387|0.5535|5.5940 |55.50|0.5346|0.5385|
> |4x|S3Diff|4.9803 |65.82|0.6361|0.6596|5.0305 |64.08|0.6328|0.6481|
> |4x|S3Diff + Ours|**4.8637**|**67.18**|**0.6459**|**0.6835**|**4.9414**|**65.31**|**0.6405**|**0.6680**|
> |16x|LIIF|11.4815|25.94|0.2860|0.3024|11.8734|26.73|0.2896|0.3178|
> |16x|SeeSR|9.6798 |41.68|0.4310|0.4669|9.8865 |40.02|0.4210|0.4601|
> |16x|S3Diff|6.7383 |51.14|0.5305|0.5886|7.2399 |50.32|0.5370|0.5841|
> |16x|S3Diff + Ours|**6.7310**|**56.82**|**0.5752**|**0.6168**|**6.8698**|**56.46**|**0.5836**|**0.6218**|
> |64x|LIIF|16.9215|20.02|0.3451|0.4131|17.4912|20.43|0.3522|0.4250|
> |64x|SeeSR|16.9095|21.83|0.4106|0.4193|16.9275|22.68|0.4167|0.4309|
> |64x|S3Diff|16.1421|21.54|0.3893|0.4917|16.5506|22.00|0.3968|0.5089|
> |64x|S3Diff + Ours|**8.7770**|**48.90**|**0.5490**|**0.5801**|**8.9794**|**48.07**|**0.5560**|**0.5909**|
> |256x|LIIF|23.8949|26.05|0.4108|0.5380|24.3300|26.01|0.4116|0.5423|
> |256x|SeeSR|20.9635|25.81|0.4438|0.5193|19.9628|25.94|0.4429|0.5203|
> |256x|S3Diff|16.7809|25.92|0.4324|0.5359|16.9952|25.91|0.4312|0.5415|
> |256x|S3Diff + Ours|**10.7668**|**43.59**|**0.5438**|**0.5570**|**10.5001**|**43.31**|**0.5417**|**0.5673**|
>
>
> ---
> **W2, Q2. [wycu] requests additional explanation of training details.**
>
> We appreciate the opportunity to further clarify the training details. Importantly, the SR model is *frozen*, and is *NOT* jointly trained with the VLM. The two are trained in a completely disjoint manner, and when applying CoZ on a pretrained off-the-shelf SR model we can bypass SR model training completely.  Additionally, when training our custom OSEDiff model (with Stable Diffusion 3.0 as backbone), DAPE is the source of caption $c$, as in the original OSEDiff paper.
>
> In a question below, the reviewer also requests clarification between random degradation and 4x specific upscaling. By *random degradation (same settings as OSEDiff)* we refer to the degradation pipeline of Real-ESRGAN, which includes 4x upscaling as inquired by the reviewer. By *4x specific upscaling* we refer to strictly creating LR images by only performing 4x downsampling of HR images with no sorts of randomness. We would like to assure the reviewer that we follow the standard procedure in the paper to ensure overall robustness of our SR model for real-world images.
>
> ---
> **W3. [wycu] requests quantitative results of different VLMs, including the critic VLM.**
>
> We would first like to remind the reviewer that a **MOS (mean-opinion-score) test** among different VLMs is available in **Section D of Supplementary Material**.  MOS testing is a widely accepted quantitative metric in perceptual super-resolution literature (e.g., FAN [3], SRGAN [4], PULSE), and it is particularly relevant in our case, as our method explicitly aims to align the VLM with **human preferences**.
>
> Furthermore, we include **additional quantitative results** to assess the quality of prompts created by different VLMs while accounting for human preference. Specifically, we include quantitative results for the metrics **ImageReward** [5] and **HPSv2** [6] to assess the quality of VLM prompts used when zooming into images at extreme magnifications. We compare the alignment between generated prompt and original input image, to accurately assess preservation of semantic consistency. The table below includes quantitative results for using the **InternVL2.5-8B** critic VLM. Interestingly, our GRPO fine-tuned VLM even outperforms the critic VLM on the ImageReward metric, highlighting the effectiveness of GRPO fine-tuning.
>
> For each row: **Single** refers to using prompts generated from only the LR input, i.e. $p_\phi(c_i|x_{i-1})$; **Multi** refers to using prompts generated from multi-scale image prompts, i.e. $p_\phi(c_i|x_{i-1},x_{i-2})$; **GRPO** refers to using $p_\phi(c_i|x_{i-1},x_{i-2})$ with GRPO fine-tuned $\phi$; **Critic** refers to using multi-scale image prompts generated by the critic VLM.
>
> Results for 64x magnification:
>
> ||DIV2K||DIV8K||
> |---|---|---|---|---|
> ||ImageReward ↑|HPSv2 ↑|ImageReward ↑|HPSv2 ↑|
> |Single|-1.9009|0.1784|-1.7699|0.1811|
> |Multi|-1.5222|0.1978|-1.3238|0.2033|
> |GRPO|-1.3558|**0.2087**|-1.2825|**0.2099**|
> |Critic|**-1.2480**|0.1900|**-1.1219**|0.1931|
>
> Results for 256x magnification:
>
> ||DIV2K||DIV8K||
> |---|---|---|---|---|
> ||ImageReward ↑|HPSv2 ↑|ImageReward ↑|HPSv2 ↑|
> |Single|-1.9705|0.1753|-1.8706|0.1777|
> |Multi|-1.7624|0.1850|-1.6635|0.1874|
> |GRPO|-1.6336|**0.2045**|-1.5634|**0.2041**|
> |Critic|**-1.4764**|0.1781|**-1.3353**|0.1816|
>
> ---
> **W4. [wycu] comments about relatively small quantitative improvement over baseline CoZ in Table 1.**
>
> As we approach larger magnifications, the quantitative improvement of using VLM prompts also increases, supporting our claim that the incorporation of VLM prompts helps *fill in* for insufficient conditioning at high magnifications. Specifically, at 256x magnification, **10.030 &rarr; 9.6405 decrease in NIQE** can be observed for DIV8K, representing a **non-negligible decrease of 4.20%**.
>
> Moreover, we would like to emphasize that incorporating VLM prompts proves to be better than null prompts **regardless of dataset or metric** at high magnifications of 64x or 256x.
> This **consistent** improvement provides strong evidence for the effectiveness and reliability of our approach.
>
> ---
> **Q1. [wycu] suggests including computational performance.**
>
> We give a detailed runtime analysis below. The average inference time required (in seconds) to apply CoZ on a single image is evaluated on 500 images of the DIV2K dataset. We divide inference into super-resolution (SR) and prompt extraction (PE), and analyze computational time required for each phase.
>
> |Scale|Phase|OneStep (DAPE)|CoZ (Null)|CoZ (DAPE)|CoZ (VLM)|
> |---|---|---|---|---|---|
> |4x|SR|0.1467|0.1443|0.1460|0.1445|
> |4x|PE|0.0136|0.0000|0.0130|0.3777|
> |16x|SR|0.1462|0.2886|0.2912|0.2896|
> |16x|PE|0.0130|0.0000|0.0254|0.7068|
> |64x|SR|0.1462|0.4329|0.4363|0.4349|
> |64x|PE|0.0131|0.0000|0.0382|1.0301|
> |256x|SR|0.1462|0.5774|0.5816|0.5802|
> |256x|PE|0.0132|0.0000|0.0509|1.3505|
>
> ---
> **Q3. [wycu] questions the rationale for using InternVL2.5-8B, and inquires of using different critic VLMs.**
>
> InternVL2.5-8B was selected as the critic VLM as a demonstration that different families of VLMs can be leveraged in our framework. To prove the robustness of our method, we additionally explore using Qwen2.5-VL-7B-Instruct as the critic VLM and present below the results for up to 5K steps. No other setting was changed except for selection of critic VLM. We present the Critic Reward and Total Reward for up to 5K steps, which are shown to consistently and significantly increase in a steep curve. Thus, we expect actual performance of the final model to likewise produce high quality results after rewards finally converge. We will include quantitative, qualitative results in our final manuscript.
>
> ||Critic Reward|Total Reward|
> |---|---|---|
> | 500|0.6391|1.0949|
> |1000|0.6704|1.1311|
> |1500|0.6958|1.1581|
> |2000|0.7186|1.1532|
> |2500|0.7324|1.1675|
> |3000|0.7666|1.2316|
> |3500|0.7849|1.2511|
> |4000|0.7944|1.2660|
> |4500|0.8221|1.2828|
> |5000|0.8483|1.3003|
>
> ---
> **Q4. [wycu] questions the captions for the first and middle row of Figure 7.**
>
> The captions for the first row are generated by a VLM from only the LR input, i.e. $p_\phi(c_i|x_{i-1})$ due to the AR-1 modeling. Captions for the middle row are generated by a VLM using multi-scale image prompts, i.e. $p_\phi(c_i|x_{i-1},x_{i-2})$ from the AR-2 modeling.
>
> ---
> **Q5. [wycu] recommends discussing works that explore the use of text in SR.**
>
> Thank you for the suggestion. We will elaborate more on related works that explore text prompt usage in SR.
>
> ---
> **Q6. [wycu] suggests explicitly indicating SR ratio in Figure 5.**
>
> We will further explain that the top two rows are of SR ratio 64x, and that the bottom two rows are of SR ratio 256x.  We will clarify this at the final version.
>
> ---
> **References**
>
> [1] Learning Continuous Image Representation with Local Implicit Image Function, CVPR 2021
>
> [2] Degradation-Guided One-Step Image Super-Resolution with Diffusion Priors
>
> [3] Progressive Face Super-Resolution via Attention to Facial Landmark, BMVC 2019
>
> [4] Photo-Realistic Single Image Super-Resolution Using a Generative Adversarial Network, CVPR 2017
>
> [5] ImageReward: Learning and Evaluating Human Preferences for Text-to-Image Generation, NeurIPS 2023
>
> [6] Human Preference Score v2: A Solid Benchmark for Evaluating Human Preferences of Text-to-Image Synthesis

---

> ### Comment · Reviewer_wycu · 2025-08-07
> **Official Comment by Reviewer wycu**
>
> Thank you to the authors for the rebuttal, which addresses most of my concerns. I appreciate the paper's motivation and presentation. I suggest incorporating the response into the revised paper. I will be increasing my rating to Accept.

---

> > ### Author Response · Authors · 2025-08-08
> > **Thanks for increasing the final rating!**
> >
> > Thank you very much for your constructive feedback and increasing your final rating. We appreciate your positive remarks, and we are glad to hear that your concerns have been resolved. If any other questions or comments arise, we would be happy to discuss them.

---

### Official Review · Reviewer_3H4j · 2025-07-03

**Clarity:** 4
**Significance:** 4
**Originality:** 4
**Rating:** 6
**Confidence:** 5

**Summary:**

This paper presents **Chain-of-Zoom (CoZ)**, a model-agnostic framework for extreme super-resolution (up to 256×) by autoregressively chaining multiple intermediate-scale inferences using a fixed 4× SR model. The method integrates multi-scale-aware prompts extracted by a vision-language model (VLM), which is further fine-tuned via GRPO-based reinforcement learning to align with human preference. Experimental results on DIV2K and DIV8K datasets show perceptual quality improvements over baselines, especially at high magnification factors.

**Questions:**

1. My theoretical analysis suggests that CoZ’s **SR model compute cost is bounded** by ~4/3× that of a one-step baseline (e.g., OSEDiff), due to decreasing image sizes at each zoom step (approximate cost: 1 + 1/4 + 1/16 + ...). However, the **VLM cost grows linearly** with zoom steps (e.g., 4× VLM calls for 256× upscaling using 4× SR). Is this analysis correct? Could the authors provide empirical runtime numbers to support this?

2. How does CoZ handle **full-image inference**? The paper only mentions cropping and resizing local patches. Is there a recommended strategy (e.g., overlapping sliding windows, global prompt sharing) for coherent full-resolution image generation?

3. Are there any failure cases where VLM prompts lead to semantic drift or hallucination over zoom steps? Could qualitative examples be added?

**Ethical Concerns:**

["NO or VERY MINOR ethics concerns only"]

**Final Justification:**

The authors have provided clear and detailed responses that address my main concerns. I appreciate their efforts and find the clarifications satisfactory. I will keep my original score.

**Limitations:**

yes

**Quality:**

4

**Strengths And Weaknesses:**

### Strengths
- **Well-motivated**: Addresses a real limitation of fixed-scale SR models at extreme upscaling factors.
- **Modular and model-agnostic**: CoZ can wrap around existing SR models without retraining them.
- **Innovative use of VLM**: Multi-scale prompt generation and GRPO-based alignment are novel and well-integrated.
- **Strong qualitative results**: Visual comparisons are convincing at 64× and 256×.
- **Comprehensive experiments**: Uses multiple no-reference metrics and ablations.

### Weaknesses

(**These are all practical concerns for deployment, but they do not detract from the fact that CoZ is an excellent and innovative contribution.**)

- **Incomplete discussion of full-image inference**: The method focuses on local patches, but lacks a clear pipeline for applying CoZ to full-resolution images.
- **VLM becomes a compute bottleneck**: Each zoom step requires a separate VLM inference, which grows linearly with the number of steps.
- **No timing or runtime analysis**: The paper presents no quantitative evaluation of inference cost compared to baseline methods.

---

> ### Author Rebuttal · Authors · 2025-07-31
>
> **W1, Q2. [3H4j] comments on a lack of discussion regarding full-image inference.**
>
> Though our original explanation of Chain-of-Zoom focuses on local patches, we would like to assure the reviewer that Chain-of-Zoom can be easily applied to super-resolution of full-resolution images. Such full-image super-resolution can be  easily done using the same tiling method as used in prior works (*e.g.*, SeeSR and OSEDiff).  Specifically, we use tiling of VAE and tiling of latents as describe in the following. More specifically,
>
> 1. Given an LR image, it is resized to the target resolution and VAE encoding is done in tiles, which can be easily performed even in settings with limited GPU memory.
> 2. The encoded (low-resolution) latent is tiled into overlapping patches. For latent sizes of $64\times64$, as is usual for input image resolutions of $512\times512$, we find that overlaps of 16 to sufficiently work well.
> 3. Each low-resolution patch of $64\times64$ passes through the super-resolution network to become high-resolution patches, each guided by patch-specific prompts generated by the prompt-extractor VLM.
> 4. The high-resolution patches are combined, multiplied by Gaussian weights in overlapping regions for smooth transposition between patches.
>
> Following the procedure above, we successfully produced full-image super-resolution results without any boundary artifacts.  We will provide detailed  procedure as well as qualitative results regarding full-image super-resolution in our final manuscript.
>
> ---
> **W3, Q1. [3H4j] suggests on giving additional runtime analysis compared to baseline methods.**
>
>  Below, we give detailed runtime analysis for each method across varying scales (4x ~ 256x). The average inference time required (in seconds) to apply CoZ on a single image is evaluated on 500 images of the DIV2K dataset. We divide inference into two phases: super-resolution (SR) and prompt extraction (PE); accurate analysis on the computational time required for each phase is given accordingly.
>
> | Scale | Phase | OneStep (DAPE) | CoZ (Null) | CoZ (DAPE) | CoZ (VLM) |
> |-------|-------|----------------|------------|------------|-----------|
> | 4x    | SR    |     0.1467     |   0.1443   |   0.1460   |   0.1445  |
> |       | PE    |     0.0136     |   0.0000   |   0.0130   |   0.3777  |
> | 16x   | SR    |     0.1462     |   0.2886   |   0.2912   |   0.2896  |
> |       | PE    |     0.0130     |   0.0000   |   0.0254   |   0.7068  |
> | 64x   | SR    |     0.1462     |   0.4329   |   0.4363   |   0.4349  |
> |       | PE    |     0.0131     |   0.0000   |   0.0382   |   1.0301  |
> | 256x  | SR    |     0.1462     |   0.5774   |   0.5816   |   0.5802  |
> |       | PE    |     0.0132     |   0.0000   |   0.0509   |   1.3505  |
>
> ---
> **W2. [3H4j] comments on the VLM becoming a computational bottleneck.**
>
> We agree that the VLM can become a computational bottleneck, especially when approaching extreme scales (*e.g.*, 64x, 256x). Computational cost for the SR model increases linearly, since the actual 4x super-resolution we achieve is fixed as (128x128 &rarr; 512x512) for each zoom step. Meanwhile, computational cost for prompt extraction also increases linearly, regardless of the prompt extractor used.  While the use of  VLM as the prompt extractor is a current computational bottleneck, we envision that further work could be done to decrease this computational bottleneck, for example by generating multiple prompts across zoom steps while using the VLM only once.
>
> ---
> **Q3. [3H4j] questions if any failure cases were observed.**
>
> When extracting text prompts, using the AR-2 modeling allows for the VLM to consider a larger receptive field compared to when using the Markov assumption. However, the VLM is still unable to consider the *full* image as recursions continue, thus out-of-context prompts can be created in cases where the global context can not be sufficiently inferred by the VLM.
>
> A specific example can be observed when zooming into an image of a jellyfish with red tentacles (the 37th image of the DIV2K dataset). As we zoom into the center of the image by $4\times$ magnification four times recursively, we get the prompts "jellyfish", "red lines", "red cables", "red wires" in their respective orders. As the VLM loses any hint of the jellyfish in its receptive field, it loses the ability to infer that the red lines are actually tentacles of a jellyfish. Thus, there is a minor *semantic drift* from what would be red tentacles of a jellyfish into red wires. To solve such issues of semantic drift, we believe that subsequent research could be done on further widening the receptive field of the VLM.
>
> We will add qualitative examples of failure cases with semantic drift (including the jellyfish example just described)  in our final manuscript.

---

> > ### Comment · Reviewer_3H4j · 2025-08-05
> > **Response to author's rebuttal**
> >
> > Thank you to the authors for the detailed clarification. My previous concerns have been sufficiently addressed. I will maintain my original score.

---

> > > ### Author Response · Authors · 2025-08-06
> > > **Thanks for the positive comments!**
> > >
> > > Thank you for revisiting our clarification and for confirming that your concerns have been resolved. We're glad to hear that you are comfortable maintaining your original score. Should any additional questions arise, we would be happy to discuss them further.

---

### Decision · Program_Chairs · 2025-09-17

**Decision:**

Accept (spotlight)

**Comment:**

This paper presents Chain-of-Zoom (CoZ), a novel autoregressive framework for extreme super-resolution (SR) that scales up to 256×. It makes a compelling and timely contribution to the SR community by tackling the longstanding challenge of scaling SR to extreme magnifications. The introduction of multi-scale VLM guidance with preference alignment is both novel and technically sound. While computational overhead remains a consideration, the method is general, effective, and well validated. After the rebuttal, reviewers reached a clear positive consensus (6554). The AC agrees with the reviewers and recommends acceptance.